# Coordinate β-adrenergic inhibition of mitochondrial activity and angiogenesis arrest tumor growth

Cristina Nuevo-Tapioles [1,2,3], Fulvio Santacatterina [1,2,3], Konstantinos Stamatakis [1], Cristina Núñez de Arenas[1,2,3], Marta Gómez de Cedrón [4], Laura Formentini [1,2,3] & José M. Cuezva [1,2,3 ✉]

Mitochondrial metabolism has emerged as a promising target against the mechanisms of tumor growth. Herein, we have screened an FDA-approved library to identify drugs that inhibit mitochondrial respiration. The β1-blocker nebivolol specifically hinders oxidative phosphorylation in cancer cells by concertedly inhibiting Complex I and ATP synthase activities. Complex I inhibition is mediated by interfering the phosphorylation of NDUFS7. Inhibition of the ATP synthase is exerted by the overexpression and binding of the ATPase Inhibitory Factor 1 (IF1) to the enzyme. Remarkably, nebivolol also arrests tumor angiogenesis by arresting endothelial cell proliferation. Altogether, targeting mitochondria and angiogenesis triggers a metabolic and oxidative stress crisis that restricts the growth of colon and breast carcinomas. Nebivolol holds great promise to be repurposed for the treatment of cancer patients.

[1] Departamento de Biología Molecular, Centro de Biología Molecular Severo Ochoa, Consejo Superior de Investigaciones Científicas-Universidad Autónoma de Madrid (CSIC-UAM), Madrid, Spain. [2] Centro de Investigación Biomédica en Red de Enfermedades Raras (CIBERER), ISCIII, Madrid, Spain. [3] Instituto de Investigación Hospital 12 de Octubre, Madrid, Spain. [4] Instituto Madrileño de Estudios Avanzados (IMDEA) Food Institute, Universidad Autónoma de Madrid, 28049 Madrid, Spain. ✉email: jmcuezva@cbm.csic.es

Cancer constitutes a major health problem worldwide. Despite the existence of standard protocols and therapies for colon[1] and breast[2] cancer, the establishment of new therapeutic approaches is imperative to minimize the social and economic burden caused by these diseases. An enhanced aerobic glycolysis is one of the hallmarks of cancer[3,4] and glycolysis itself has been proposed as a potential chemotherapeutic target to combat the disease[5–7]. However, the complete understanding of the metabolic dependencies of tumors could provide additional strategies to restrain tumor growth[7]. In this regard, mitochondrial metabolism also affords a promising target to fight cancer progression[8–11].

Mitochondria play essential cellular functions regulating the provision of metabolic energy by oxidative phosphorylation (OXPHOS), the execution of cell death, and intracellular signaling by $Ca^{2+}$ and reactive oxygen species (ROS)[12–14]. In OXPHOS, the ATP synthase catalyzes the synthesis of ATP using as driving force the proton electrochemical gradient generated by the respiratory chain[12]. The activity of the ATP synthase is regulated by its physiological inhibitor, the ATPase inhibitory factor 1 (IF1), a small nuclear-encoded mitochondrial protein that is highly overexpressed in some human carcinomas[15–17]. The activity of IF1 as an inhibitor of the ATP synthase is regulated under normal physiological conditions by its expression and by the phosphorylation of S39 through the activity of a mitochondrial protein kinase A-like activity[18]. In human carcinomas, IF1 is found predominantly dephosphorylated and hence acting as an inhibitor of the ATP synthase[18], contributing to metabolic reprogramming of the cells to an enhanced glycolytic phenotype[15,16,19]. Hence, the IF1/ATP synthase system offers a potential therapeutic target in cancer and other human disorders[20], as recently stressed in aging and dementia[21].

Cancer drug discovery and development is a costly and lengthy task that spans more than a decade before the drug is ready for treating patients[22]. The fact that only a few drugs are finally approved for use puts pressure on their price to compensate the investment in drug discovery. Drug repurposing has emerged as an alternative strategy to overcome the costs and time invested in cancer drug discovery[22]. The success of drug repurposing relies on that the compounds have already been introduced for another indication and tested in human therapy with acceptable known side effects that improve the quality of life of the patients[22].

Herein, we screen an FDA-approved library of small compounds to find drugs that could inhibit the activity of the mitochondrial ATP synthase in cancer cells and consequently could prevent tumor growth. We find 13 compounds that inhibit mitochondrial respiration and the activity of the ATP synthase. We study in detail the mechanisms by which the third-generation β1-blocker nebivolol halts colon and breast tumor growth in vivo. The results emphasize the relevance of blocking β1-adrenergic signaling to inhibit cancer progression, supporting the repurposing of nebivolol as an anticancer drug to be used in combined chemotherapy of the oncologic patient.

## Results

**Nebivolol inhibits mitochondrial respiration**. To identify the inhibitors of OXPHOS that could interfere with cancer progression, we screened an FDA-approved library of 1018 small compounds that in short-term treatment of 3 h affected mitochondrial respiration of HCT116 colon cancer cells (Fig. 1a). The study was initially carried out in a Seahorse XFe96 analyzer using the oligomycin-sensitive respiration (OSR) as a reporter of the drug's effect because it represents an estimate of the activity of the ATP synthase. We identified compounds that enhanced or inhibited OSR by 40% when compared to cells treated with the vehicle (dimethyl sulfoxide (DMSO)) (Fig. 1a). Further in-depth investigation of the effect of the inhibitors on OXPHOS was carried out using the Seahorse XF24.

Thirteen FDA-approved drugs significantly inhibited basal, maximum mitochondrial respiration, and OSR of HCT116 cells (Fig. 1b and Table 1). Blocking of cardiac β-adrenoceptors by propranolol recently showed the relevance of the PKA/cAMP signaling pathway in preventing the phosphorylation of IF1 and the subsequent inhibition of OXPHOS in heart mitochondria[18]. Hence, of the 13 inhibitors of respiration identified (Table 1), we focused on nebivolol for further in-depth study because it is a β1-adrenergic inhibitor whose mechanism of action is compatible with targeting OXPHOS, both at the level of the respiratory chain[23,24] and at the level of the ATP synthase[18].

Nebivolol inhibited mitochondrial respiration of both colon HCT116 (Fig. 1c) and breast MDA-MB-231 (Fig. 1d) cancer cells when glucose (Fig. 1c, d) or palmitate (Supplementary Fig. 1a) were used as respiratory substrates. Titration of the effect of increasing concentrations of nebivolol in OSR revealed an $IC_{50}$ of ~0.9 and ~2.1 μM in HCT116 and MDA-MB-213, respectively (Supplementary Fig. 1b). Similar results were obtained for the $IC_{50}$ of nebivolol on the maximum respiratory rate (Supplementary Fig. 1b). Moreover, nebivolol also inhibited mitochondrial respiration of neuroblastoma (SH-SY5Y), lung (A549), and ovarian (OVCAR8) cancer cells (Fig. 1e). Remarkably, nebivolol did not affect mitochondrial respiration of the Hs 578T normal breast cell line (Fig. 1f) nor of mouse primary neuronal cultures and C2C12 myocytes (Fig. 1g). The lack of effect of the drug on mitochondrial respiration in isolated liver organelles (Supplementary Fig. 1c) excluded the possibility of a direct inhibitory effect of nebivolol in mitochondria.

The effect of four additional β1-blockers, bisoprolol, metoprolol, betaxolol, and acetobutolol, also significantly inhibited the mitochondrial respiration of HCT116 cancer cells (Fig. 1h). Interestingly, ICI 118,551 and SR 59230A, respectively, representing a β2- and β3-adrenergic receptor blockers, did not affect mitochondrial respiration in HCT116 cancer cells (Fig. 1i). These results suggest that the inhibition of mitochondrial respiration in cancer cells stems from β1-adrenergic blockade. In fact, only cells that responded to nebivolol express β1-adrenergic receptors (Fig. 1j).

**Nebivolol inhibits mitochondrial ATP synthesis**. Treatment of colon and breast cancer cells with nebivolol significantly diminished the synthesis of ATP by mitochondrial ATP synthase as assessed in permeabilized colon and breast cancer cells (Fig. 2a). In response to nebivolol, cancer cells partially induced aerobic glycolysis as a result of the inhibition of ATP supply by OXPHOS (Fig. 2b). In agreement with the inhibition of mitochondrial respiration by nebivolol, the drug triggered a slight but significant increase in mitochondrial membrane potential (ΔΨm) in cancer cells (Fig. 2c). Interestingly, and consistent with the inhibition of the ATP synthase by nebivolol, oligomycin, an inhibitor of the ATP synthase, exerted a similar increase in ΔΨm in both cancer cells (Fig. 2c). Moreover, we also observed a slight but significant increase in cellular ROS levels in nebivolol-treated cells when compared to controls (Fig. 2d). However, nebivolol-treated cells did not show significant differences in cellular proliferation (Supplementary Fig. 2a) and cell death responses to different death-inducing agents (Supplementary Fig. 2b).

**Nebivolol increases IF1 expression**. Interestingly, the effect of nebivolol in cellular respiration occurred in both cell lines in the absence of changes in the expression of subunits of respiratory complexes, albeit for the sharp increased expression of IF1

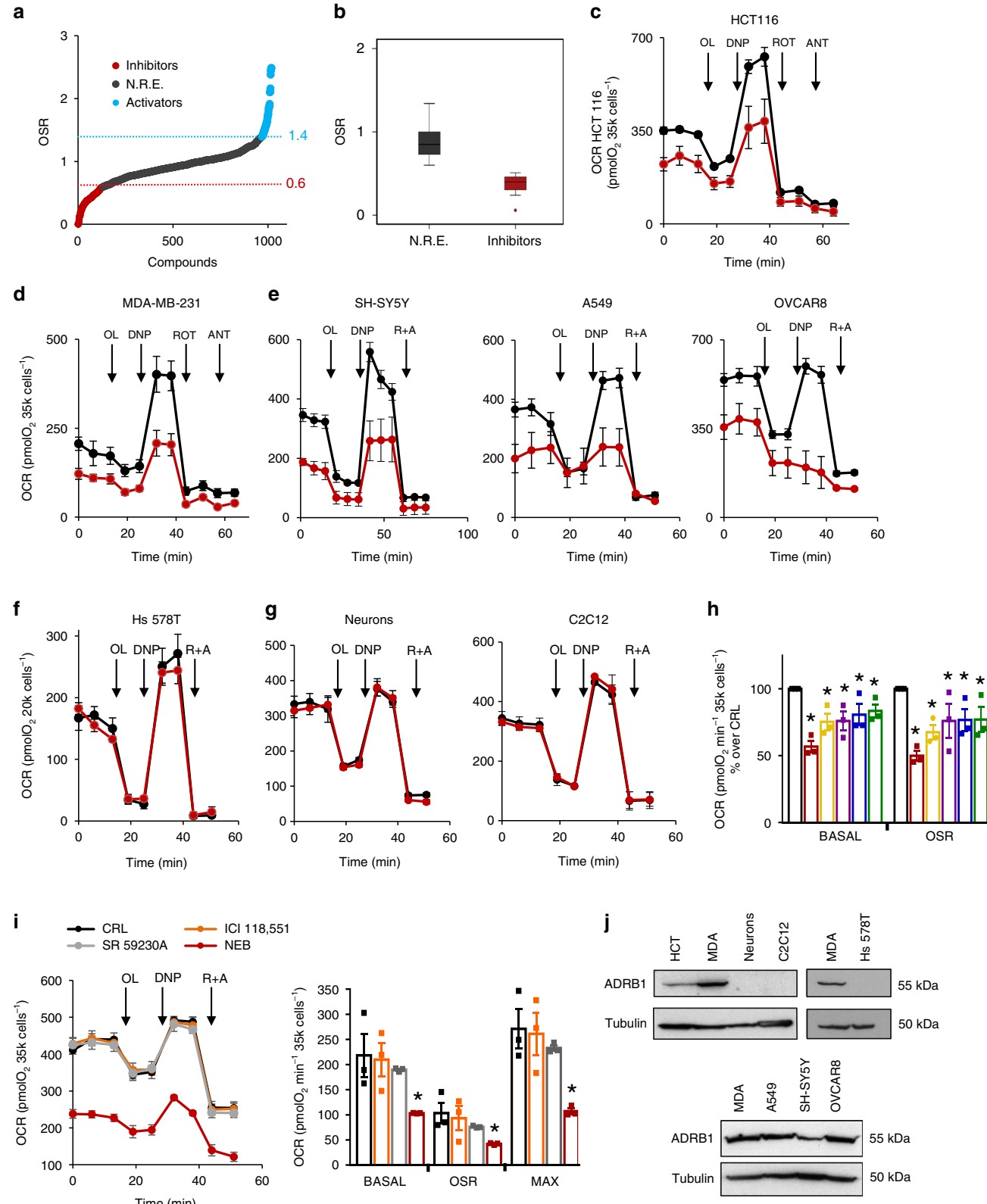

(Fig. 2e). The nebivolol-mediated accumulation of IF1 is independent of significant changes in IF1-mRNA abundance (Supplementary Fig. 2c) and its accumulation could explain the inhibition of ATP synthesis observed in the cells (Fig. 2a).

**The organization of OXPHOS complexes is not affected**. Rapid changes in mitochondrial respiratory activity could also be related

to a different organization of OXPHOS complexes[25,26]. However, BN-PAGE analysis of the supramolecular organization of OXPHOS complexes after nebivolol treatment of HCT116 cells indicated no relevant changes in their supramolecular organization (Fig. 3a), supporting that the inhibition of maximum respiration triggered by nebivolol should be ascribed to another mechanism. Interestingly, IF1 was also found bound to the ATP

**Fig. 1 Nebivolol inhibits mitochondrial respiration in cancer cells. a** Effect of 1018 FDA-approved drugs on the oligomycin-sensitive respiration (OSR) of HCT116 colon cancer cells. Activators (≥1.4; in blue) and inhibitors (≤0.6; in red) that affected OSR by 40% when compared to control are highlighted. Compounds with no relevant effect (NRE) are shown in gray. **b** Box plots represent 25th to 75th percentiles with the median value in the middle line, with all data represented from minimal to maximal values indicated with whiskers of the inhibitors (red box) or the compounds with no relevant effect (NRE, black box). **c** Respiratory profile of HCT116 cells treated (red trace) or not (black trace) for 3 h with nebivolol (1 μM). OCR oxygen consumption rate, OL oligomycin, DNP 2,4-dinitrophenol, ROT rotenone, ANT antimycin A. **d** Respiratory profile of MDA-MB-231 cells treated (red trace) or not (black trace) for 3 h with nebivolol (1 μM). **e** Respiratory profile of neuroblastoma SH-SY5Y, lung A549 and ovarian OVCAR8 cells treated (red trace) or not (black trace) for 3 h with nebivolol (1 μM). **f** Respiratory profile of breast Hs 578T cells treated (red trace) or not (black trace) for 3 h with nebivolol (1 μM). **g** Respiratory profile of mouse primary neuronal cultures (left panel) and C2C12 myoblast (right panel) treated (red trace) or not (black trace) for 3 h with nebivolol (1 μM). **h** Basal and OSR of HCT116 cells treated for 3 h with nebivolol (red bar, *$p = 0.01$ and 0.01), bisoprolol (yellow bar, *$p = 0.01$ and 0.01), metoprolol (purple bar, *$p = 0.04$ and 0.03), betaxolol (blue bar, *$p = 0.02$ and 0.05), acebutolol (green bar, *$p = 0.03$ and 0.02) (1 μM) or left untreated (black bar). **i** Respiratory profile of HCT116 cells treated for 3 h (1 μM) with β1-antagonist nebivolol (NEB, red trace and bar; Basal: *$p = 0.05$; OSR: *$p = 0.04$; MAX: *$p = 0.01$), β2-antagonist ICI 118,552 (orange trace and bar), β3-antagonist SR 59230A (gray trace and bar) or left untreated (CRL, black trace and closed bar). **j** Representative western blots of the expression of β1-adrenergic receptor (ADRB1) in cancer cells (HCT116, MDA-MB-231, A549, SH-SY5Y, and OVCAR8) and normal cells (Hs 578T, primary neurons and C2C12). Tubulin is shown as loading control. Bars indicate the mean ± SEM of three biological replicates. *$p < 0.05$ and **b**–**e** $p < 0.05$ when compared to CRL by two-sided Student's $t$ test. See also Supplementary Fig. 1. Source data are provided as a Source Data file.

---

**Table 1 Potent inhibitors of mitochondrial respiration.**

| Compound | Indication | Target | OSR | Basal | Pubmed hits | Clinical trials |
|---|---|---|---|---|---|---|
| Sorafenib | Cancer | VEGFR, PDGFR | 0.33 ± 0.10 | 0.28 ± 0.09 | 1/1 | 31/25 |
| Regorafenib | Cancer | VEGFR | 0.46 ± 0.01 | 0.67 ± 0.05 | 1/0 | 1/72 |
| Ponatinib | Cancer | Abl | 0.46 ± 0.08 | 0.60 ± 0.04 | 0/0 | 0/0 |
| Itraconazole | Cancer | P450 | 0.27 ± 0.10 | 0.17 ± 0.10 | 1/0 | 3/0 |
| Telaprevir | Infection | HCV Protease | 0.39 ± 0.10 | 0.58 ± 0.08 | 0/0 | 0/0 |
| Sulfameter | Infection | Dihydropteroate synthase | 0.39 ± 0.07 | 0.50 ± 0.05 | 0/0 | 0/0 |
| Crystal violet | Infection | Thioredoxin reductase 2 | 0.06 ± 0.09 | 0.26 ± 0.10 | 0/0 | 0/0 |
| Butoconazole | Infection | Steroid synthesis | 0.47 ± 0.10 | 0.55 ± 0.09 | 0/0 | 0/0 |
| Nebivolol | Cardiovascular | β1-adrenergic receptor | 0.28 ± 0.10 | 0.45 ± 0.08 | 0/0 | 0/0 |
| Pazopanib | Cardiovascular | VEGFR | 0.40 ± 0.10 | 0.47 ± 0.05 | 0/0 | 17/4 |
| Lomerizine | Cardiovascular | Ca$^{2+}$ channel | 0.51 ± 0.01 | 0.51 ± 0.03 | 0/0 | 0/0 |
| Glyburide | Endocrinology | β cells | 0.40 ± 0.09 | 0.50 ± 0.10 | 0/0 | 0/0 |
| Cimetidine | Inflammation | Histamine H2 receptor | 0.24 ± 0.02 | 0.41 ± 0.04 | 0/0 | 0/0 |

Compounds of the library that inhibit more than 40% mitochondrial respiration of HCT116 colon cancer cells. Name of the drug, clinical indication, target, OSR, and basal respiration. Values are mean ± SEM of five experiments referred to control values. $p < 0.05$ when compared to CRL by two-sided Student's $t$ test. PubMed search results for searching each drug with either "breast cancer" (numerator) or "colon cancer" (denominator) AND mitochondria. Search results for the clinical trials (clinicaltrial.gov) for breast/colon cancer.

---

synthase and other oligomeric states of the enzyme in human cancer cells (Fig. 3a), in agreement with recent findings in mouse tissues[27]. Moreover, we observed that the amount of IF1 co-fractionating with F1-ATPase and the monomeric ATP synthase (Complex V) in BN-PAGE increased significantly (Fig. 3b), mimicking the increase in IF1 observed in cellular extracts after nebivolol treatment (Fig. 2e).

**Nebivolol affects phosphorylation of OXPHOS complexes.** It is known that the phosphorylation of serine residues regulates the activity of proteins of OXPHOS[18,23,24]. In this regard, the inhibition of maximum respiration induced by treatment with nebivolol could be due to a post-transcriptional β-adrenergic blockade of the respiratory chain (Fig. 1c–e) and of the ATP synthase (Fig. 2a). Hence, we investigated the existence of modifications in serine-phosphorylated proteins in OXPHOS complexes by BN-PAGE in response to nebivolol treatment (Fig. 4a). A ~50% decrease in phosphorylation of proteins contained in super-complexes (SC) was observed when the cells were treated with nebivolol (Figs. 3a, 4a for comparison). Interestingly, protein phosphorylation in another complex migrating at ~600 kDa was unaltered in the same situation (Fig. 4a).

Since IF1 is also found in high molecular mass complexes co-migrating with SC (Fig. 3a), we initially studied if the reduction in protein phosphorylation by nebivolol at the level of SC (Fig. 4a) could be ascribed to deficient phosphorylation

of IF1. Immunoprecipitation of IF1 and blotting with anti-phosphoserine antibody revealed that most of IF1 in HCT116 cells is in the dephosphorylated state (Fig. 4b) as compared to cells treated with the membrane permeable db-cAMP (Fig. 4b). To quantitate the relative amount of dephospho-IF1, we carried out 2D-gels to distinguish the phosphorylated forms of IF1 by differences in their pI (Fig. 4c)[18]. Consistent with immunoprecipitation experiments (Fig. 4b), we observed that HCT116 and MDA-MB-231 cells contained most of IF1 dephosphorylated (85–100%) focusing at pI 8 (Fig. 4c), which corresponds to the migration of the S39A phosphodeficient IF1 mutant protein (red dotted line in Fig. 4c). Only a small amount of phosphorylated IF1 (~pI 7.2) was found in HCT116 cells corresponding to the migration of the phosphomimetic S39E-IF1 mutant (blue dotted line in Fig. 4c). Overall, the results suggest that inhibition of ATP synthesis by nebivolol is not due to a relevant change in the phosphorylation of IF1 and most likely results from its increased expression in response to the 3-h treatment (Fig. 2e).

**The upregulation of IF1 by nebivolol inhibits ATP synthase.** To illustrate this latter possibility, we studied the ATP hydrolase activity of the ATP synthase in isolated mitochondria from both HCT116 and MDA-MB-231 cancer cells (Fig. 4d). The results confirmed that the activity of the enzyme was significantly inhibited in nebivolol-treated cells (Fig. 4d). Moreover, we found

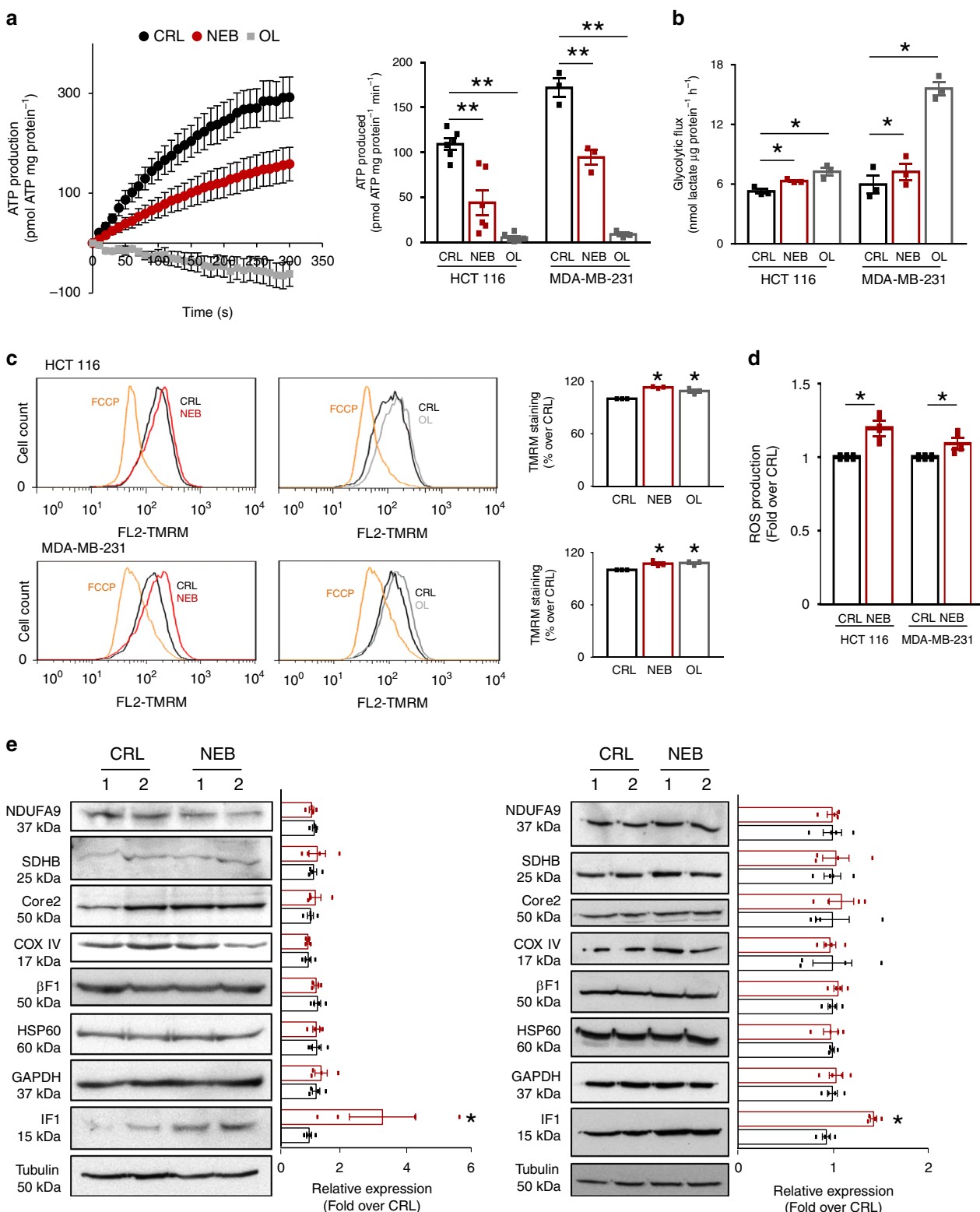

that the mitochondrial content of IF1 was significantly augmented in nebivolol-treated cells (Fig. 4e), in agreement with the increased IF1 found co-migrating with Complex V and F1-ATPase in BN-gels (Fig. 3a, b). Consistently, silencing of IF1 expression in HCT116 (Fig. 4f) and MDA-MB-231 (Supplementary Fig. 3a) cells abolished the effect of nebivolol on basal and OSR of the cells. However, the effect of nebivolol on the maximum respiratory rate was maintained in IF1-silenced cells (Fig. 4f and Supplementary

Fig. 3a), indicating that in addition to the effect of the drug on the ATP synthase (Figs. 2a, 4d, f), nebivolol was also affecting the activity of some of the respiratory complexes.

**Nebivolol prevents phosphorylation and inhibits complex I.** To verify this idea, we determined the activity of complexes I, II, and IV of the respiratory chain (Fig. 4g–i). Nebivolol significantly diminished the activity of complex I (Fig. 4g) without affecting

**Fig. 2 Nebivolol inhibits mitochondrial ATP synthase.** HCT116 and MDA-MB-231 cells were treated during 3 h with 1 μM nebivolol (NEB; red dots and bars), oligomycin (OL; gray dots and bars) or left untreated (CRL; closed dots or bars). **a** Left panel, kinetic representation of the production of ATP in digitonin-permeabilized cells. The inhibition of ATP synthase activity was accomplished by the addition of 30 μM OL. Right panel, histograms show the ATP synthetic activity. Six replicates of six (HCT116, **$p = 0.001$ and **$p = 0.00001$) and three (MDA-MB-231, **$p = 0.004$ and **$p = 0001$) different biological samples are shown. **b** Glycolytic flux measured by the initial rates of lactate production. Three replicates of three different biological HCT116 (*$p = 0.01$ and 0.01) and MDA-MB-231 (*$p = 0.04$ and $p = 0.01$) samples are shown. **c** Plots and histograms of TMRM$^+$ stained cells to assess the mitochondria membrane potential (ΔΨm). FCCP (orange) collapses ΔΨm. Three replicates of three different biological HCT116 (*$p = 0.01$ and 0.01) and MDA-MB-231 (*$p = 0.05$ and 0.02) samples are shown. See Supplementary Fig. 7a for gating strategy. **d** Histograms show ROS production in HCT116 (*$p = 0.02$) and MDA-MB-231 (*$p = 0.05$) cells treated (red) or not (black) for 3 h with 1 μM nebivolol. Three replicates of three different biological samples are shown. See Supplementary Fig. 7b for gating strategy. **e** Representative western blots of four different biological samples of the expression of mitochondrial proteins from complex I (NDUFA9), complex II (SDHB), complex III (Core 2), complex IV (COX IV), complex V (β-F1-ATPase) and IF1 (*$p = 0.011$ and 0.02), Hsp60 and GAPDH and tubulin as loading controls. Two different samples of control (CRL; closed bars) and nebivolol-treated (NEB; red bars) cells are shown. Bars indicate the mean ± SEM of the indicated samples. *$p < 0.05$ and **$p < 0.01$ when compared to CRL by two-sided Student's $t$ test. See also Supplementary Fig. 2. Source data are provided as a Source Data file.

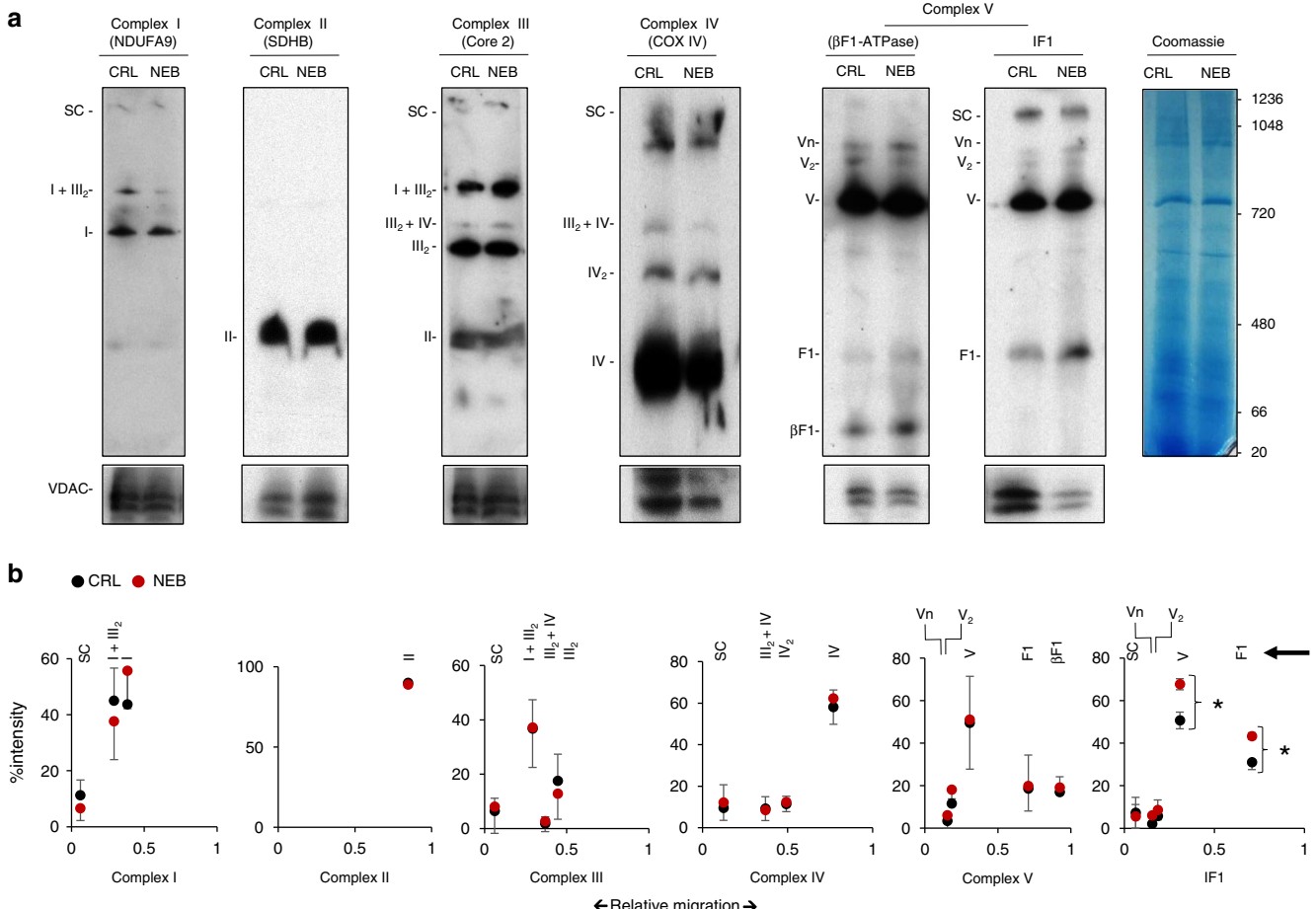

**Fig. 3 Nebivolol increases the amount of IF1 bound to the ATP synthase.** HCT116 cells were treated (NEB, red) or not (CRL, black) during 3 h with 1 μM nebivolol. **a** Representative BN-immunoblots of mitochondrial membrane proteins blotted with the antibodies of the indicated subunits of the different OXPHOS complexes. The migration of supercomplexes (SC); complex I and I + III$_2$ (NDUFA9); complex II (SDHB); complex III$_2$, III$_2$ + IV and I + III$_2$ (Core 2); complex IV, IV$_2$ and III$_2$ + IV (COX IV); oligomers (Vn), dimers (V$_2$), monomers (V), F1-ATPase and β-F1-ATPase subunit of the ATP synthase and their co-migration with IF1 are indicated. The migration of molecular mass markers is indicated in the Coomassie-stained gel and applied for all blots. VDAC and Coomassie-stained gel are shown as loading controls. **b** Plots of the quantification of the relative migration (arrowhead) of fractionated OXPHOS complexes. Bars indicate the mean of three different experiments ±SEM. *$p = 0.042$ (V) and 0.039 (F1) when compared to CRL by two-sided Student's $t$ test. Source data are provided as a Source Data file.

the activity of the other complexes (Fig. 4h, i). Changes in the activity of Complex I correlate with differences in the phosphorylation of subunits of the complex[23]. Nebivolol also significantly diminished the phosphorylation of SC (Fig. 4a), where complex I is usually present (Fig. 3a). Consistently, nebivolol specifically inhibited the phosphorylation of complex I

immunocaptured proteins from HCT116 and MDA-MB-231 cells (Fig. 4j). Phosphoproteomic analysis by MS-spectrometry of the peptides derived from the 15–20 kDa region of the gels (see red box in Fig. 4j) indicated that S117 contained in a tryptic peptide derived from NDUFS7 subunit was the only peptide from complex I that was not phosphorylated when the cells were treated

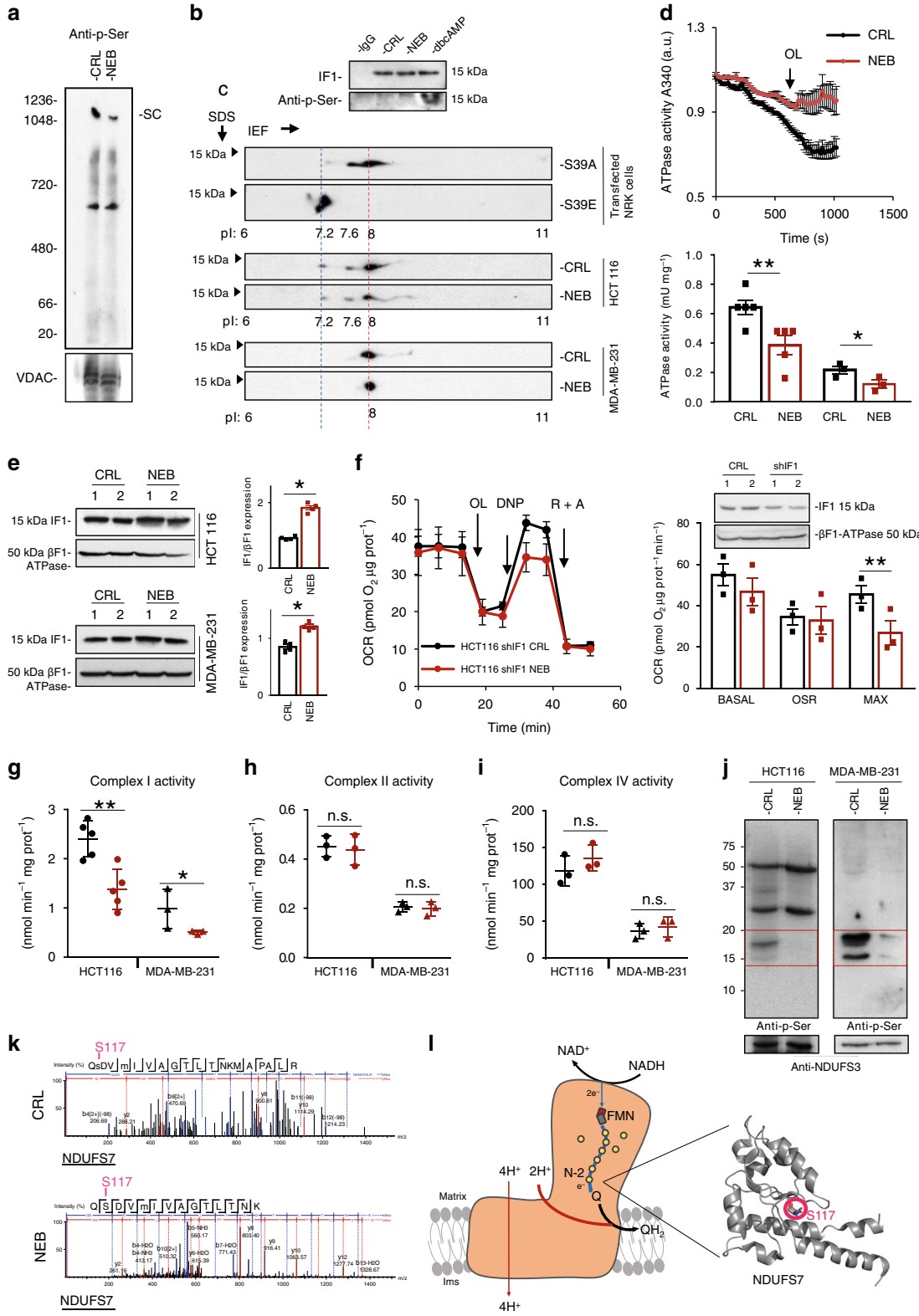

with nebivolol (Fig. 4k). Interestingly, this subunit is located in the ubiquinone binding site of complex I (Fig. 4l).

**Nebivolol delays the in vivo growth of colon carcinomas.** Although nebivolol treatment does not affect the proliferation and death of cancer cells growing in culture (Supplementary Fig. 2a, b), we next tested whether restraining OXPHOS by inhibiting complex I and complex V activities with nebivolol could impede colon and breast cancer progression in vivo. For this purpose, nude mice were subcutaneously injected with HCT116-Luc cells into the right and left flanks. When the tumors had reached a volume of ~100 mm³, mice were treated with daily doses of nebivolol 5 days a week. A control NaCl-treated group was also included for comparison. Within 6 days after initiation of nebivolol treatment, mice revealed a significant reduction in tumor luminescence as compared to the controls (Fig. 5a) and

**Fig. 4 Nebivolol inhibits Complex I and prevents phosphorylation of NDUFS7.** HCT116 and MDA-MB-231 cells were treated (NEB, red dots and bars) or not (CRL, closed dots and bars) during 3 h with 1 μM nebivolol. **a** Representative BN-immunoblot of mitochondrial membrane proteins blotted with the anti-p-Ser antibody. The migration of supercomplex (SC) and molecular mass markers is indicated. VDAC is shown as loading control. The BN was repeated at least three times with similar results. **b** IF1 immunoprecipitated from HCT116 cells treated as indicated. Cells were also treated with 100 μM db-cAMP. Anti-IF1 and anti-phosphoserine blots are shown. Nonspecific immunoglobulin G (IgG) was included as control. The IP was repeated at least three times with similar results. **c** Blots showing fractionated cellular proteins on 2D-gels and blotted against IF1. The isoelectrofocusing (IEF) of IF1-mutants (S39A and S39E) expressed in NRK cells. Representative blots of the endogenous IF1 in HCT116 and MDA-MB-231 cells treated (NEB) or not (CRL) with nebivolol. The pIs (under the blots) and MW are indicated. The 2D-gels have been repeated at least three times with similar results. **d** Determination of the hydrolytic activity of the ATP synthase in isolated mitochondria from HCT116 and MDA-MB-231 cells. Inhibition of the activity was accomplished by the addition of 30 μM OL. Histograms show the ATP hydrolytic activity in six replicates of five (HCT116, $**p = 0.009$) and three (MDA-MB-231, $*p = 0.048$) different biological samples. **e** Representative western blots of isolated mitochondria from HCT116 ($*p = 0.017$) and MDA-MB-231 ($*p = 0.02$) cells. Histograms show the expression of IF1 relative to βF1-ATPase subunit of the ATP synthase of four different biological samples. **f** Respiratory profile of IF1-silenced HCT116 cells treated (HCT116 shIF1 NEB, red trace) or not (HCT116 shIF1 CRL, black trace) with nebivolol. Basal, oligomycin-sensitive respiration (OSR, $**p = 0.002$) and maximum (MAX) respiration of three biological replicates are shown in the histograms. OCR oxygen consumption rate, OL oligomycin, DNP 2,4-dinitrophenol, R rotenone, A antimycin A. Representative western blot of control HCT116 and IF1-silenced cells are included. The expression of IF1 and βF1-ATPase subunit of the ATP synthase is shown. **g-i** The histograms show the enzymatic activity of Complex I ($n = 5$; $**p = 0.003$ and $*p = 0.01$), II ($n = 3$), and IV ($n = 3$) in isolated mitochondria. **j** Immunocapture of Complex I blotted with anti-phosphoserine antibody. NDUFS3 is used as loading control. The red square identifies the region of the gel used for phosphoproteomic identification. Immunocapture was repeated three times with similar results. **k** Mass spectrum of the tryptic peptides derived from NDUFS7 of cells treated or not with nebivolol. pS117 phosphopeptide (CRL, upper panel) and dephospho S117 peptide (NEB, lower panel) are shown. **l** Schematic of Complex I structure shows the flux of electrons from NADH down to ubiquinone (Q). The FMN, Fe-S clusters (N-2 and yellow circles), and approximate location of NDUFS7 are highlighted. The direction of proton pumping from the matrix to the intermembrane space (ims) is indicated. The crystallographic structure of human NDUFS7 is shown indicating the location of S117. Image taken from ref. [70] (PDB: 5XTD). Bars indicate the mean ± SEM of different experiments as indicated. $*p < 0.05$ and $**p < 0.01$ when compared to CRL by two-sided Student's $t$ test. See also Supplementary Fig. 3a. Source data are provided as a Source Data file.

significantly arrested tumor growth thereafter (Fig. 5b). Kaplan−Meier survival analysis showed that nebivolol treatment increased the lifespan of mice as compared to NaCl-treated controls (Fig. 5c). The restraining of tumor growth in nebivolol-treated mice resulted from a significant inhibition of cellular proliferation, as revealed by Ki67 staining (Fig. 5d), and an enhanced cell death, as revealed by the activation of caspase-3 (Fig. 5e). Moreover, tumors of nebivolol-treated mice showed a sharp reduction of ATP content (Fig. 5f).

Since nebivolol treatment slightly increased ROS in breast and colon cancer cells (Fig. 2d), we next studied the potential influence of nebivolol in affecting the redox status of the carcinomas. A significant increase in nitrotyrosine-modified proteins—a modification related to protein inactivation by reactive nitrogen species—was found in tumors of nebivolol-treated mice when compared to controls (Supplementary Fig. 3b). Moreover, the expression of mitochondrial proteins involved in the antioxidant response, such as the mitochondrial ROS scavenging enzymes superoxide dismutase 2 (SOD2), peroxiredoxine 3 (PRx3), and glutathione reductase (GR), was significantly increased in tumors of mice treated with nebivolol (Fig. 5g). In contrast, the expression of glucose-6-phosphate dehydrogenase (G6PDH), peroxiredoxine 6 (PRx6), and catalase (Cat) was either nonaffected or significantly diminished in the tumors of nebivolol-treated mice (Fig. 5g). Interestingly, and as shown previously in cells (Fig. 2e) and in mitochondria (Fig. 4e), tumors of nebivolol-treated mice showed a significant increase in IF1 expression (Fig. 5g). Altogether, the results indicate that inhibition of mitochondrial respiration in colon carcinomas of nebivolol-treated mice results in a metabolic and redox crisis as demonstrated by the diminished tumor ATP content (Fig. 5f), protein oxidative damage (Supplementary Fig. 3b), and increased mitochondrial antioxidant response (Fig. 5g) when compared to carcinomas of control mice.

To assess the effectiveness of nebivolol in combined therapy of colon cancer, we studied its effect in combination with 5-fluouracil (5FU). Interestingly, the combined treatment (NEB + 5FU) significantly reduced tumor volume after 6 days of treatment (Fig. 5h). Moreover, Kaplan−Meier survival curves showed that NEB + 5FU

significantly increased the lifespan of mice when compared to control or 5FU-treated mice (Fig. 5i).

**β1-adrenergic blockade prevents tumor angiogenesis.** In order to explain the different cytotoxic effect of nebivolol between in vitro and in vivo studies, we first address the possibility that a metabolite of the degradation pathway of nebivolol could be involved in cytotoxicity. 4OH-nebivolol, the major secondary metabolite of nebivolol degradation, has no effect on mitochondrial respiration (Supplementary Fig. 4a) and cellular proliferation (Supplementary Fig. 4b). Moreover, 4OH-nebivolol showed no cytotoxicity in colon (Supplementary Fig. 5a) and breast (Supplementary Fig. 5b) cancer cells. Likewise, the response of cancer cells to death induced by staurosporine (Supplementary Fig. 5a) or tamoxifen (Supplementary Fig. 5b) was not significantly affected by 4OH-nebivolol, emphasizing the role of blocking β1-receptors to arrest tumor growth in vivo.

β-adrenergic signaling is involved in angiogenesis in vivo[28]. Hence, we next assessed the potential implication of tumor angiogenesis as a contributing factor that could explain the different cytotoxic effect of nebivolol between in vitro and in vivo data. Analysis of the expression of the angiogenic marker isolectin B4 (IB4) in endothelial cells of the carcinomas suggested that nebivolol arrested microvessels formation (Fig. 5j). The inhibition of angiogenesis in carcinomas of nebivolol-treated mice was additionally confirmed by the reduced expression of the endothelial cell CD31 marker (Fig. 5k), the basement membrane marker laminin (Fig. 5l) and of αSMA, a marker of pericytes (Fig. 5m). Altogether, this indicates that nebivolol triggers a significant inhibition of tumor angiogenesis (Fig. 5j–m).

**Nebivolol arrests endothelial cells proliferation.** Interestingly, whereas the expression of VEGF was not affected in nebivolol-treated carcinomas (Fig. 6a), nebivolol significantly diminished the expression of VEGF-receptor2 (VEGFR2) (Fig. 6a). Analysis of VEGFR2 expression in human umbilical vein endothelial cells (HUVEC) treated with nebivolol showed no relevant differences

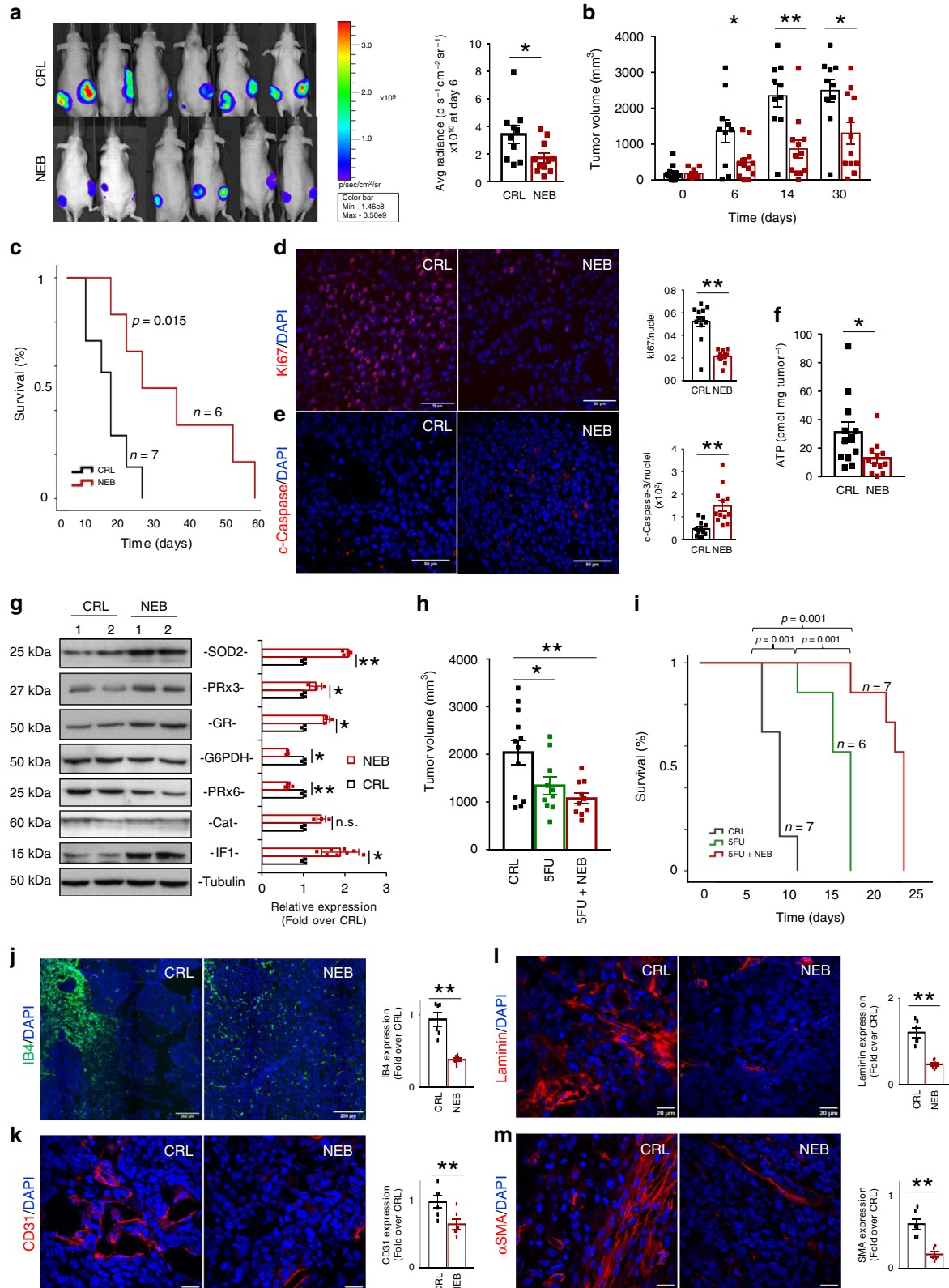

in VEGFR2 expression when compared to controls (Fig. 6b). However, nebivolol significantly reduced HUVEC cell number (Fig. 6c) by inhibiting its proliferation (Fig. 6d). Thus, further indicating that the reduction of VEGFR2 expression (Fig. 6a) in nebivolol-treated carcinomas results from a limited angiogenesis.

It has been estimated that most ATP requirements in endothelial cells are provided by glycolysis[29]. Consistently, whereas nebivolol

had no relevant effect on mitochondrial respiration of HUVEC cells (Supplementary Fig. 6a), despite expressing β1-adrenergic receptor (Fig. 6e), nebivolol significantly inhibited glycolysis in these cells (Fig. 6f). In fact, nebivolol significantly arrested cell cycle progression (Fig. 6g, see also Supplementary Fig. 6b) by preventing the activation of ERK (Fig. 6h), which is known to block cell cycle in S phase[30], arresting HUVEC cells at G0/G1 (Fig. 6g, see also

**Fig. 5 Nebivolol halts colon cancer growth in mice.** HCT116-Luc cells were injected into the flanks of nude mice. Mice were treated with saline (CRL; black trace and bars; $n = 7$), 10 mg kg$^{-1}$ nebivolol (NEB; red trace and bars; $n = 6$), 0.2 mg kg$^{-1}$ 5-fluorouracil (5FU, green trace and bar; $n = 6$) or both compounds (NEB + 5FU, red trace and red bar; $n = 7$) and sacrificed when tumor volume reached 2000 mm$^3$. **a** Left panel shows representative images of the bioluminescence of HCT116-Luc tumors in mice after 6 days of initiation of the indicated treatment. Right panel, quantification of light emission of the cells (CRL, $n = 10$; NEB $n = 11$ tumors; *$p = 0.02$). **b** Tumor volume (mm$^3$) at day 0 and after 6 (*$p = 0.02$), 14 (**$p = 0.002$) and 30 (*$p = 0.02$) days of treatment (CRL, $n = 12$; NEB $n = 12$ tumors). **c** Kaplan–Meier survival analysis. The log-rank test $p$ value (0.015) is shown. **d**, **e** Immunofluorescence microscopy images of Ki67 (red) (**d**) or cleaved active caspase-3 (c-caspase-3, red) (**e**) and DAPI (blue)-stained carcinomas treated as indicated. Scale bar, 50 μm. Histograms represent the ratio of Ki67 (**$p = 0.0001$) or c-caspase-3 (**$p = 0.0002$) positive cells relative to cell nuclei of six different biological samples. **f** ATP content of the carcinomas (CRL, $n = 12$; NEB $n = 12$ tumors; *$p = 0.05$). **g** Representative western blots of two different samples and quantification of four different samples (histograms) of the expression of SOD2 (**$p = 0.0005$), PRx3 (*$p = 0.04$), GR (*$p = 0.03$), G6PDH (*$p = 0.01$), PRx6 (**$p = 0.0001$), catalase (Cat) and IF1 (*$p = 0.05$) in tumors. Tubulin is shown as loading control. **h** Tumor volume (mm$^3$) at day 6 of the indicated treatments (CRL, $n = 12$; NEB, $n = 10$; NEB + 5FU, $n = 10$ tumors) (*$p = 0.04$ and **$p = 0.004$). **i** Kaplan–Meier survival analysis. The log-rank test $p$ value is shown. **j** Immunofluorescence microscopy images of isolectin B4 (IB4, green; **$p = 0.0002$) and DAPI (blue)-stained carcinomas treated as indicated. Scale bar, 200 μm. Histograms represent the relative expression of IB4-positive cells relative to cell nuclei of six different biological samples. **k**–**m** Immunofluorescence microscopy images of CD31 (**$p = 0.007$) (**k**), laminin (**$p = 0.00001$) (**l**) and αSMA (**$p = 0.0004$) (**m**) (red) and DAPI (blue)-stained carcinomas treated as indicated. Scale bar, 20 μm. Histograms represent the relative expression of CD31, laminin or αSMA-positive cells relative to cell nuclei of six different biological samples. Bars indicate the mean of indicated samples ± SEM. *$p < 0.05$ and **$p < 0.01$ when compared to CRL by two-sided Student's $t$ test. See also Supplementary Figs. 3–5. Source data are provided as a Source Data file.

Supplementary Fig. 6b). Overall, these findings support that nebivolol hinders tumor angiogenesis by inhibiting endothelial cell proliferation in vitro and in vivo.

**Nebivolol prevents the growth of orthotopic colon carcinomas.** Immunocompromised mice were injected with HCT116-luc cells in the cecum. After bioluminescence detection of a stable signal, mice were randomly allocated into the group of treated mice, with daily doses of nebivolol 5 days a week (NEB), and the control NaCl-treated group (CRL) (Fig. 7a, upper panel). Tumor growth was followed by the increase in bioluminescence signal every 2 days/week. Within 25 days after initiation of the treatment, mice revealed a significant reduction in tumor luminescence when compared to controls (Fig. 7a, lower panel). Nebivolol significantly decreased tumor growth by 15 days of treatment reaching a fivefold decrease at 34 days (Fig. 7b). Furthermore, nebivolol-treated mice significant developed less (Fig. 7c, left panel) and smaller (Fig. 7c, right panel) tumors when compared to NaCl-treated mice. In addition, nebivolol treatment significantly diminished the number of micrometastasis (Fig. 7d, left panel) and the macroscopic evidence of tumor angiogenesis (Fig. 7d, right panel) when compared to controls. Altogether, our results support that nebivolol also arrests tumor growth in vivo in the colon microenvironment.

**Nebivolol delays in vivo growth of breast carcinomas.** The effect of nebivolol was also assessed in breast cancer growth in vivo using an MDA-MB-231 xenograft mouse model (Fig. 8). Nebivolol treatment significantly reduced tumor volume when compared to control NaCl-treated mice (Fig. 8a). Kaplan—Meier survival curves showed that nebivolol significantly increased the lifespan of mice when compared to control (Fig. 8b). Restraining tumor growth by nebivolol treatment resulted in significant inhibition of cellular proliferation (Fig. 8c) and an enhanced cell death (Fig. 8d) in the carcinomas.

Tumors of nebivolol-treated mice showed a significant reduction in the total content of ATP (Fig. 8e). A significant increase in nitrotyrosine-modified proteins was also observed in breast carcinomas of nebivolol-treated mice (Fig. 8f). Likewise, nebivolol also increased the expression of proteins of the antioxidant response such as peroxiredoxin 3, glutathione reductase, and glucose-6 phosphate dehydrogenase, when compared to control NaCl-treated mice (Fig. 8g). In agreement with previous observations in breast cancer cells (Fig. 2e) and in isolated mitochondria (Fig. 4e), tumors

of nebivolol-treated mice revealed an increased expression of IF1 when compared to tumors of control mice (Fig. 8g). Moreover, nebivolol significantly inhibited the expression of the angiogenic markers IB4 (Fig. 8h), CD31 (Fig. 8i), Laminin (Fig. 8j), αSMA (Fig. 8k), and VEGFR2 (Fig. 8l) in breast carcinomas, further supporting a relevant role for β-adrenergic signaling in favoring tumor angiogenesis.

Overall, we show that nebivolol, by blocking β-adrenergic receptors, arrests the growth of carcinomas through the coordinate action of preventing vascularization of the tumor and inhibiting the bioenergetic function of cancer mitochondria (Fig. 9).

## Discussion

Drug repurposing offers a valuable approach to reduce the socio-economic burden of cancer[10,31,32]. Although cancer cells repro-gram their energy metabolism to an enhanced glycolysis[3,4], PGC-1-driven activation of mitochondrial OXPHOS is required for metastasis[33]. Hence, mitochondrial metabolism provides a hopeful target to fight against cancer[8–10,34]. Indeed, arsenic trioxide, an inhibitor of mitochondrial respiration, has been approved for the treatment of relapsed acute promyelocytic leukemia[35]. With this idea in mind, we screened an FDA-approved library of small compounds that, in a short-term incubation treatment, trigger a sharp inhibition of mitochondrial OXPHOS in different cancer cells. Thirteen drugs originally purposed for the treatment of cancer, infection, cardiovascular disease, endocrinology, and inflammation met the stringent selection criteria established in the screening. β-adrenergic activation of the cAMP/PKA signaling pathway is known to stimulate OXPHOS by promoting the phosphorylation of proteins of the respiratory chain and of the inhibitor of the ATP synthase[23,24,36]. Hence, we further addressed the role and mechanism of action of nebivolol, a third-generation β1-blocker[37], as a potential anticancer drug by its putative capacity to restrain the activity of OXPHOS.

We show that nebivolol inhibits mitochondrial respiration and the synthesis of ATP in a large number of cancer cells (Fig. 9). Remarkably, nebivolol has no effect in isolated mitochondria or in nontumor cells, which emphasizes its specificity and excludes any antimitotic toxicity. Recently, Gboxin, a small molecule that accumulates in cancer mitochondria driven by the membrane potential, impedes ATP synthesis and inhibits the growth of glioblastoma xenograft[10]. The inhibition of respiration mediated by nebivolol is linked to a sharp decrease in the activity of Complex I of the respiratory chain (Fig. 9), in agreement with the role of the cAMP/PKA signaling pathway in controlling

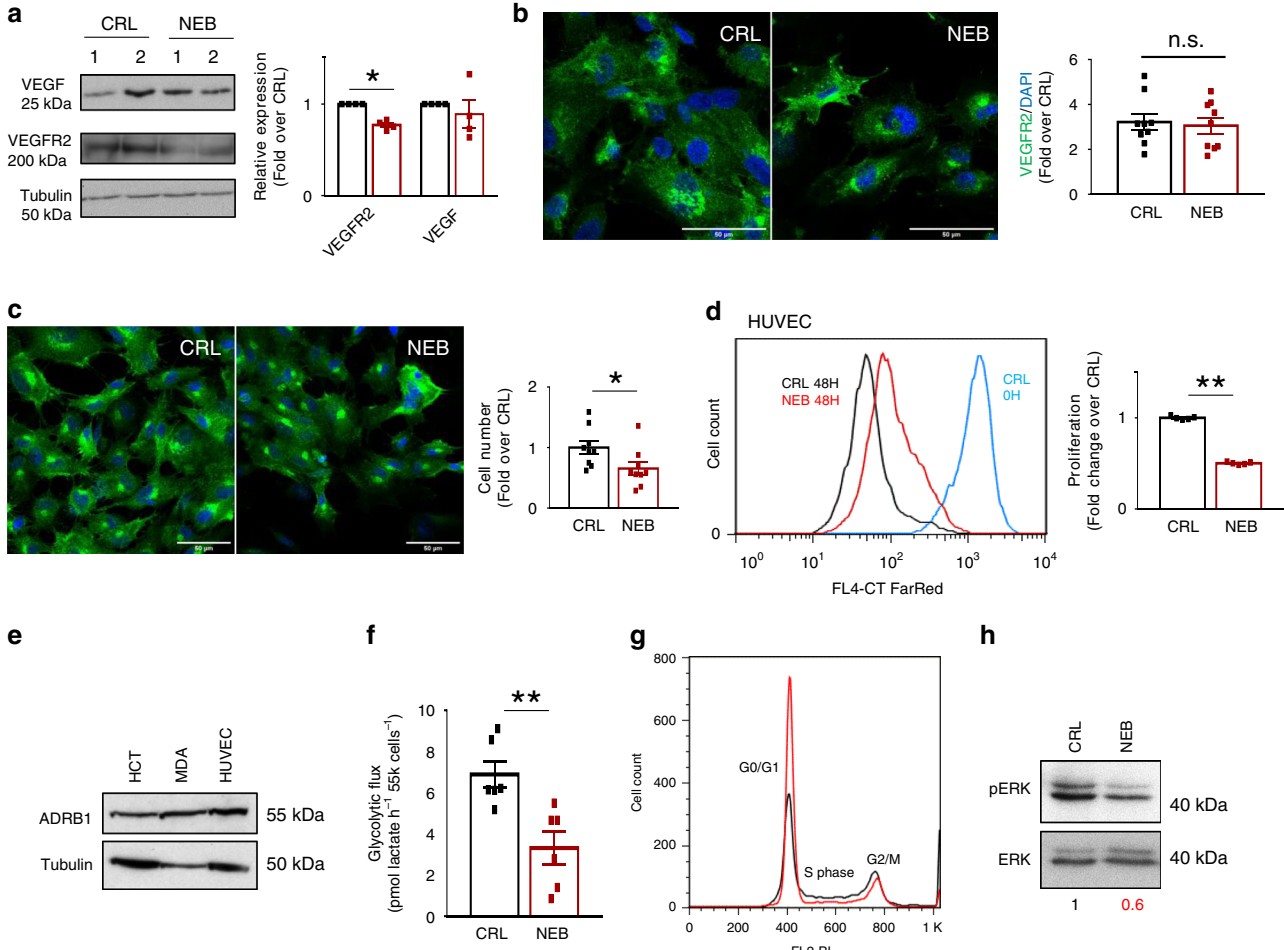

**Fig. 6 Nebivolol inhibits endothelial cells metabolism and proliferation. a** Representative western blots of two different samples and quantification of four different samples (histograms) of the expression of VEGF and VEGFR2 (*$p = 0.01$) in tumors of control (CRL) and nebivolol-treated (NEB) colon xenograft mice. Tubulin is shown as loading control. **b–h** HUVEC cells were treated (NEB, red dots and bars) or not (CRL, closed dots and bars) during 3 or 48 h with 10 μM nebivolol. **b** Immunofluorescence microscopy images of VEGFR2 (VEGFR2, green) and DAPI (blue)-stained HUVEC treated as indicated. Scale bar, 50 μm. Histograms represent the relative expression of VEGFR2-positive cells relative to cell nuclei from nine different slices. **c** Immunofluorescence microscopy images of VEGFR2 (VEGFR2, green) and DAPI (blue)-stained HUVEC treated as indicated. Scale bar, 50 μm. Histograms represent the number of cells from nine different slices. *$p = 0.03$. **d** The figure shows the proliferation after 48 h of CellTrace Red Far incorporation. The blue curve shows time 0. The plot shows the quantification of cellular proliferation of four different samples. **$p = 3.1E{-}10$. See Supplementary Fig. 7c for gating strategy. **e** Representative western blot of the expression of β1-adrenergic receptor (ADRB1) in cancer and in human umbilical vein endothelial cells (HUVEC). Tubulin is shown as loading control. Western blots were repeated three times with similar results. **f** Glycolytic flux measured by the initial rates of lactate production of six different biological samples are shown. **$p = 0.005$. **g** Plot shows the percentage of cells in G0/G1, S and G2/M phases of the cell cycle from three different samples. See Supplementary Fig. 7d for gating strategy. **h** Representative western blot of the expression of ERK and its phosphorylated form (pERK). Western blots were repeated two times with similar results. Bars indicate the mean of the indicated samples ± SEM. *$p < 0.05$ and **$p < 0.01$ when compared to CRL by two-sided Student's $t$ test. See also Supplementary Fig. 6. Source data are provided as a Source Data file.

mitochondrial respiration[23,38,39]. Metformin and its analogs, used to treat type 2 diabetes, also inhibit complex I of the respiratory chain[9,40,41] and have anticancer properties[9,42]. The exact mechanism by which metformin inhibits complex I remains unknown. However, it is interesting to note that biguanides inhibit ubiquinone reduction and stimulate ROS production by FMN at complex I[40]. The likely mechanism of action of nebivolol on complex I activity is by preventing the phosphorylation of S117 of NDUFS7[43] (Fig. 9), which is a subunit of the complex that directly couples electron transfer between the iron-sulfur cluster N2 and ubiquinone (Fig. 4l). Interestingly, as nebivolol triggered nitrotyrosine modification of cellular proteins and the induction of a mitochondrial antioxidant response, both indicative of mitochondrial ROS generation, it is tempting to suggest

that deficient phosphorylation of NDUFS7 could mediate the production of ROS in this situation (Fig. 9).

Moreover, just as metformin inhibits the ATP synthase[40], so does nebivolol, which explains the sharp reduction of ATP observed in the carcinomas of nebivolol-treated mice (Fig. 9). The inhibition of ATP synthesis by nebivolol is unrelated to the changes in the phosphorylation status of IF1[18], and that correlates with the rapid and sharp nebivolol-triggered increase in the expression of IF1 (Fig. 9). The overexpression of IF1 is already known to inhibit ATP synthesis in cancer cells[15,16,19] and in different tissues of transgenic mice that overexpress the protein in vivo[44–47]. Consistently, we show that nebivolol-treated cells have reduced ATP hydrolase activity and increased dephosphorylated IF1 bound to the ATP synthase (Fig. 9).

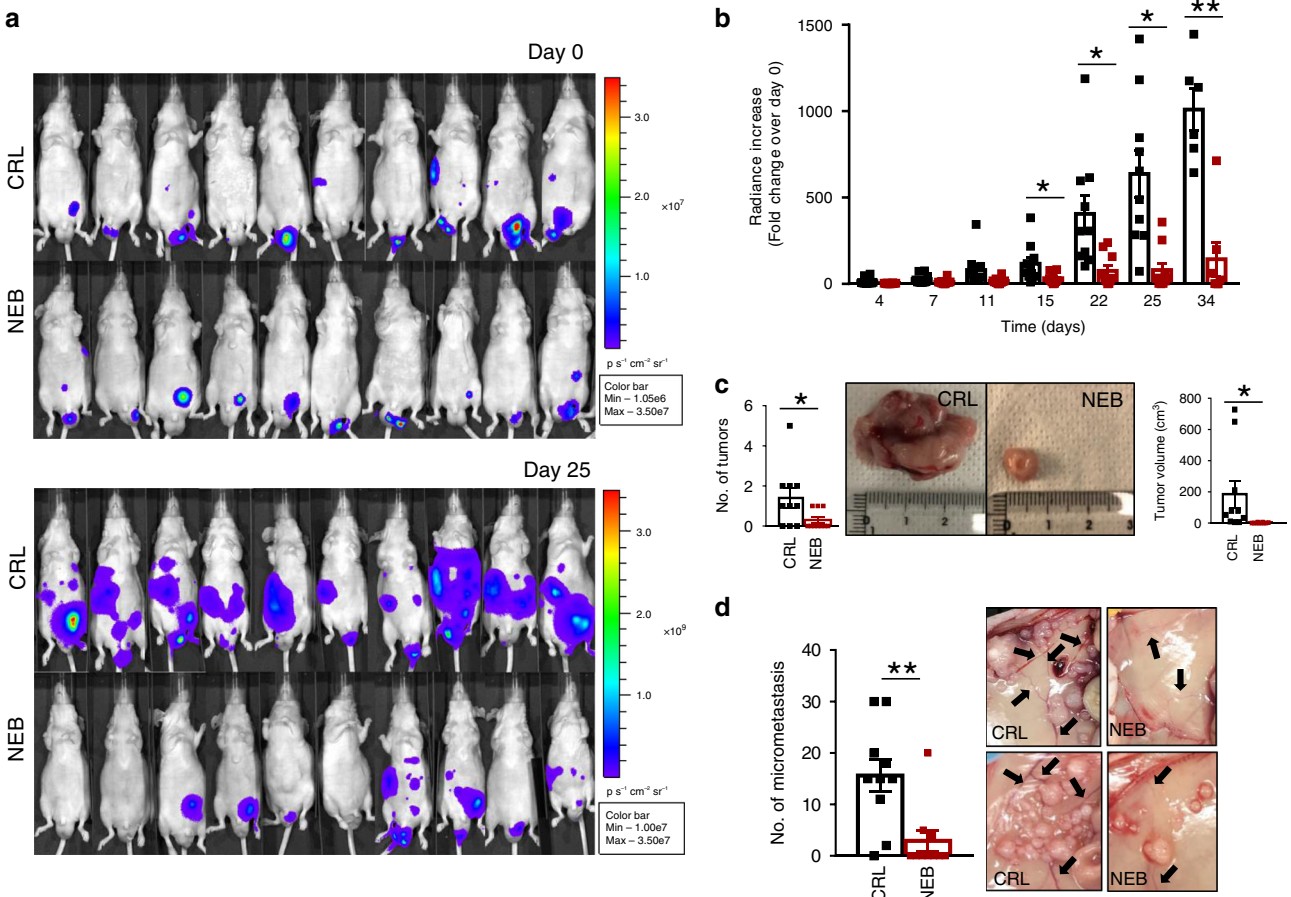

**Fig. 7 Nebivolol halts cancer growth in orthotopic colon mice.** HCT116-Luc cells were injected into the cecum of nude mice. Mice were treated with saline (CRL; black trace and bars; $n = 10$) or 10 mg kg$^{-1}$ nebivolol (NEB; red trace and bars; $n = 10$) and sacrificed after 35 days of treatment. Bars indicate the mean ± SEM. *$p < 0.05$ and **$p < 0.01$ when compared to CRL by two-sided Student's $t$ test. **a** Representative images of the bioluminescence of HCT116-Luc tumors in mice at day 0 and after 25 days of initiation of the indicated treatment. Light emission of the cells was measured after injection of 150 mg kg$^{-1}$ of D-Luciferin. **b** The histograms show the fold change of the radiance increase after 4, 7, 11, 15 (*$p = 0.03$), 22 (*$p = 0.04$), 25 (*$p = 0.02$) and 34 (**$p = 0.0002$) days of treatment. **c** Left panel shows the histograms for the quantification of the number of tumors found in the mice (*$p = 0.04$). Right panel shows representative images of the size and aspect of the tumors and the histograms for the quantification of tumor volume (*$p = 0.04$). **d** Left panel shows the histograms for the quantification of the micrometastasis found in the mice. Right panel shows representative images of the micrometastasis. Black arrows point to the vessels around the tumors (**$p = 0.003$). Source data are provided as a Source Data file.

Remarkably, the overexpression of IF1 in hepatocarcinomas[48], gastric[49], lung[50] and bladder[51] carcinomas and gliomas[52] identifies patients with worst prognosis because IF1 favors proliferation and metastatic disease. In contrast, the overexpression of IF1 in colon and breast carcinomas correlates with better patients' prognosis[16,53], stressing the importance and tissue-specific relevance of IF1 as a biomarker and target of cancer chemotherapy. Interestingly, the expression of IF1 in normal human tissues[27] and carcinomas[16] occurs independently of changes in the tissue availability of IF1 mRNA. IF1 is a mitochondrial protein with a very short half-life (2–3 h) in differentiated osteocytes and in human stem and colon cancer cells[16,54]. Consistent with these observations, the nebivolol-promoted increase in IF1 expression observed in cancer cells is unrelated to changes in IF1 mRNA abundance, supporting the idea that the β-blocker is affecting the turnover rate of the protein. Altogether, these findings emphasize the need for future studies aimed at characterizing the tissue-specific mechanisms that control IF1 expression for the prominent role it plays in regulating the bioenergetics of cancer cells and the metastatic behavior of the carcinomas.

It is interesting to note the different cell-death behavior of cells growing in culture and in vivo towards nebivolol. Remarkably,

the cancer cells expressing the β1-adrenergic receptor respond to different β1-antagonists by inhibiting mitochondrial respiration while they do not respond to β2- and β3-receptor inhibitors, supporting the role of β1-adrenergic signaling in arresting tumor growth in vivo. Tumor angiogenesis and cancer progression requires β-adrenergic signaling[28]. In fact, the overexpression of PKA is considered a hallmark that correlates with bad clinical prognosis and pathological features of the carcinomas[55]. Moreover, PKA is also involved in uncontrolled proliferation, cytoskeleton remodeling, and the migration of cancer cells[55,56] and, blocking PKA activation is known to halt cancer progression[57,58]. On the other hand, despite endothelial cells express β1-adrenergic receptors, nebivolol significantly diminished their glycolytic flux —a main pathway in endothelial cells[29]—further preventing their proliferation and tumor angiogenesis. Hence, we suggest that the differences in nebivolol cytotoxicity between in vitro and in vivo studies result from the restriction of oxygen and nutrients imposed by the deficient tumor angiogenesis in nebivolol-treated mice. Limiting vasculogenesis and mitochondrial function—the latter affecting both ATP production and the concurrent generation of mitochondrial oxidative stress at the level of complex I —are convergent pathways by which nebivolol arrests the

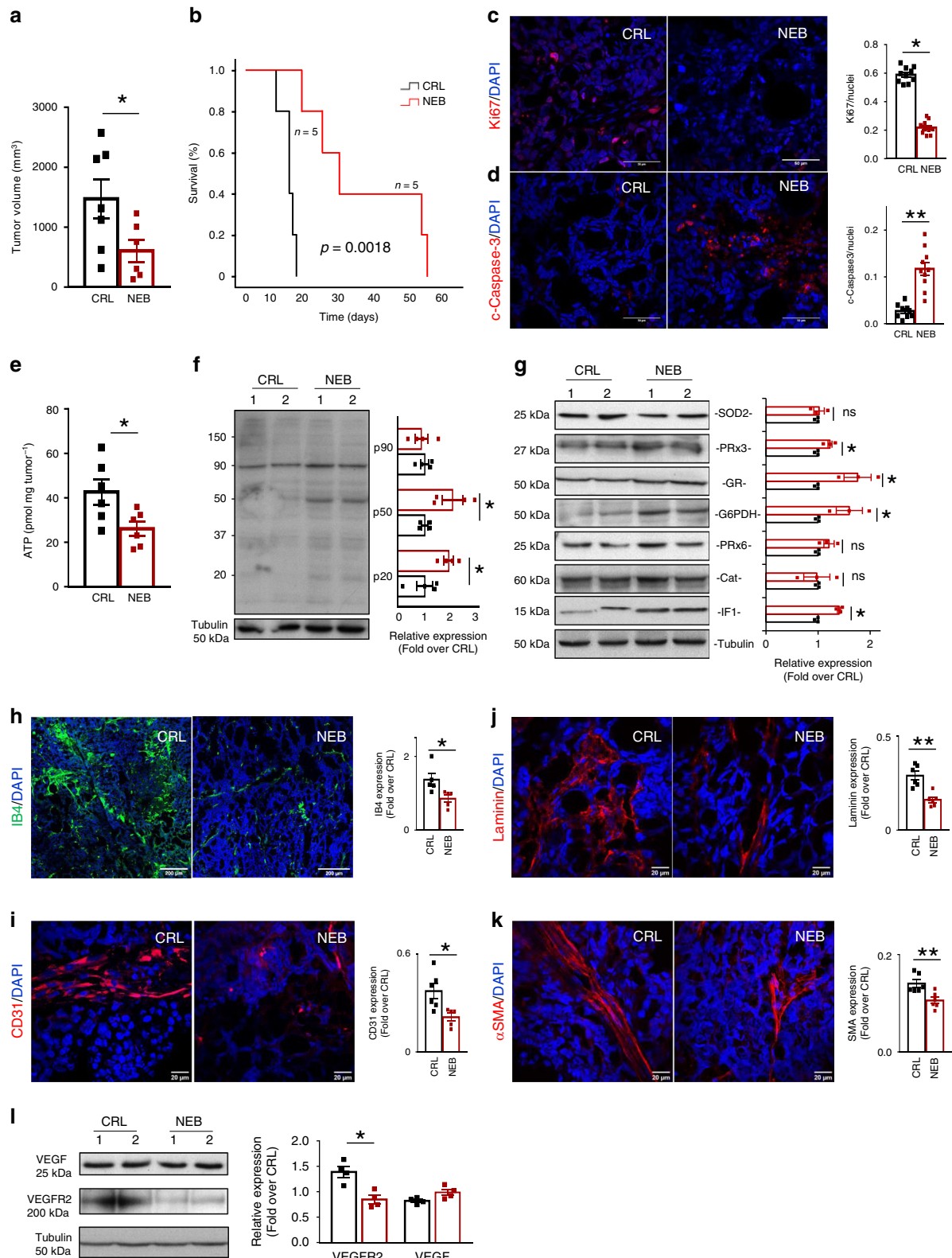

proliferation and enhances death of cancer cells growing in vivo (Fig. 9). Remarkably, in mice bearing colon carcinomas, life expectancy increased further when nebivolol was used in combination with 5FU, illustrating nebivolol's ability to potentiate the activity of the classical anticancer colorectal drug.

Basket trials are defined as those including cancer patients with carcinomas from different tissue origins sharing a common mutation and/or biomarker[59]. The expression of β1-receptors could be considered a biomarker of different cancer cells, as shown in this study and elsewhere[60]. Interestingly, there are no clinical trials in which nebivolol has been used as an anticancer agent (Table 1). Therefore, our findings point out that nebivolol is a promising drug to be repurposed to treat cancer patients in combined therapy because targeting β1-adrenergic signaling with

**Fig. 8 Nebivolol halts breast cancer growth in mice.** MDA-MB-231 cells were injected into the flanks of nude mice. Mice were treated with saline (CRL; black trace and bars; $n = 5$) or 10 mg kg$^{-1}$ nebivolol (NEB; red trace and bars; $n = 5$) and sacrificed when tumor volume reached 2000 mm$^3$. **a** Tumor volume (mm$^3$) at day 6 after treatment initiation (CRL, $n = 7$; NEB $n = 6$ tumors; *$p = 0.048$). **b** Kaplan–Meier survival analysis. The log-rank test $p$ value (0.0018) is shown. **c, d** Immunofluorescence microscopy images of Ki67 (*$p = 0.03$) (red) (**c**) or cleaved active caspase-3 (**$p = 0.009$) (c-caspase-3, red) (**d**) and DAPI (blue)-stained carcinomas treated as indicated. Scale bar, 50 μm. Histograms represent the ratio of Ki67-or c-caspase-3-positive cells relative to cell nuclei of six different biological samples. **e** ATP content of the carcinomas (CRL, $n = 6$; NEB, $n = 6$ tumors) (*$p = 0.03$). **f** Representative western blots of two different samples and quantification of $n = 4$ tumors (histograms) of nitrotyrosine modified tumor proteins. Tubulin is shown as loading control (*$p = 0.04$ and 0.03). **g** Representative western blots of two different samples and quantification of $n = 4$ tumors (histograms) of the expression of SOD2, PRx3 (*$p = 0.04$), GR (*$p = 0.04$), G6PDH (*$p = 0.05$), PRx6, catalase (Cat) and IF1 (*$p = 0.04$) in tumors. Tubulin is shown as loading control. **h** Immunofluorescence microscopy images of isolectin B4 (IB4, green) and DAPI (blue)-stained carcinomas treated as indicated. Scale bar, 200 μm. Histograms represent the relative expression of IB4-positive cells relative to cell nuclei of six different biological samples (*$p = 0.03$). **i−k** Immunofluorescence microscopy images of CD31 (*$p = 0.01$) (**i**), laminin (**$p = 0.0008$) (**j**) and αSMA (**$p = 0.006$) (**k**) (red) and DAPI (blue)-stained carcinomas treated as indicated. Scale bar, 20 μm. Histograms represent the relative expression of CD31, laminin or αSMA-positive cells relative to cell nuclei of six different biological samples. **l** Representative western blots of two different samples and quantification of four different samples of the expression of VEGF and VEGFR2. Tubulin is shown as loading control (*$p = 0.04$). Bars indicate the mean of the indicated samples ± SEM. *$p < 0.05$ and **$p < 0.01$ when compared to CRL by two-sided Student's $t$ test. Source data are provided as a Source Data file.

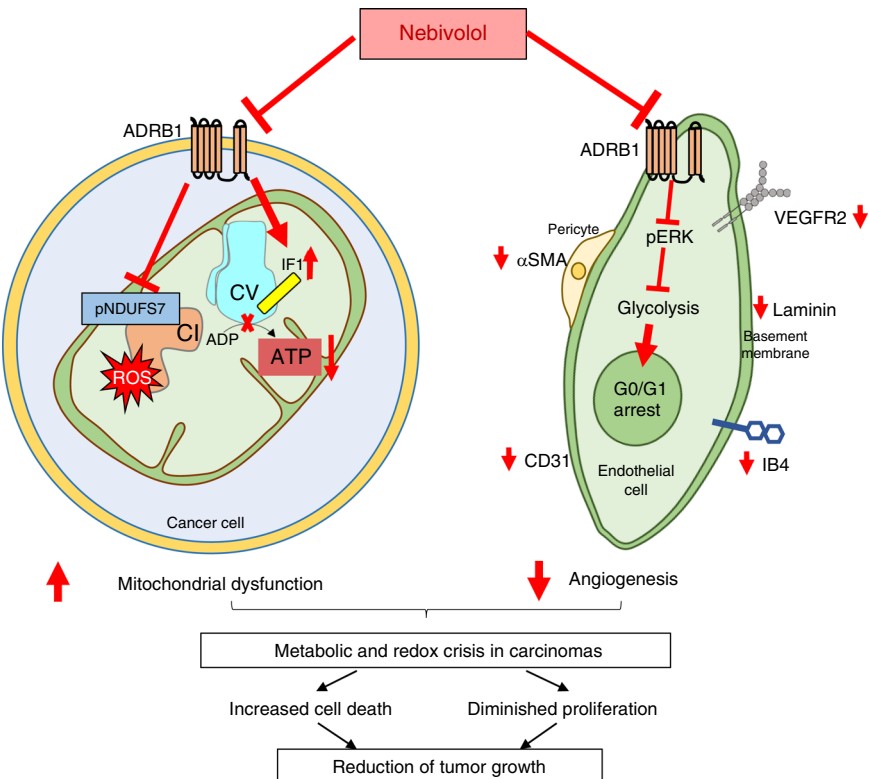

**Fig. 9 Nebivolol induces a metabolic and redox crisis in carcinomas.** The schematic summarizes the main events triggered by nebivolol to prevent the growth of colon and breast carcinomas. Nebivolol inhibits β1-adrenergic signaling in cancer cells, preventing the phosphorylation of NDUFS7 (pNDUFS7, blue rectangle in CI) that limits the activity of Complex I (CI in orange) and mitochondrial respiration, favoring the generation of ROS (red star). Concurrently, nebivolol increases the mitochondrial content of the ATPase Inhibitory Factor 1 (IF1, yellow cylinder) that binds the ATP synthase (CV in light blue) and limits ATP production in the carcinomas. Moreover, nebivolol leads to diminished tumor angiogenesis by inhibiting proliferation in endothelial cells through β1-adrenergic-mediated glycolysis inhibition and thus, cell cycle arrest. These events result in a metabolic (less ATP) and redox crisis (increased ROS) that limit cellular proliferation and enhanced death of cancer cells preventing the in vivo growth of the carcinomas.

the FDA-approved β-blocker restrains the growth of colon and breast carcinomas. In addition, the side effects of nebivolol in hypertensive patients are tolerated. Overall, incorporating nebivolol as an additional cancer drug tackles simultaneously tumor angiogenesis and restrains mitochondrial functions, resulting in an overall metabolic and oxidative stress condition that halts tumor growth.

## Methods

**Animal studies.** For the in vivo studies, 6-week-old male or female nude mice with a body weight of 30–35 g were implanted with HCT116-Luc or MDA-MB-231,

respectively. The Ethics Committee of Animal Experimentation (CSIC-UAM, CM PROEX 023/14) and the Institutional Review Board of UAM (CEI 75-1365) approved the project. Mice were housed in the Animal Facility of the CBMSO with a 12-h light/12-h dark cycle and temperatures of 18–23 °C with 40–60% humidity. Approximately, $4 \times 10^6$ HCT116-luc or MDA-MB-231 cells were injected subcutaneously into the left and the right flanks of mice to develop the xenograft model. To develop the orthotopic model, $5 \times 10^5$ HCT116-luc cells were injected between the mucosa and the muscularis externa layers of the cecal wall of anesthetized mice (isoflurane gas, 1.5%, Abbott) in 10 μL of saline solution using a 30-gauge needle. Tumor growth was monitored by bioluminescence acquisition on anesthetized mice (isoflurane gas, 1.5%, Abbott) using the IVIS Lumina II in vivo imaging system (Caliper Life Sciences) for the HCT116-Luc cells after intraperitoneal injection of 150 mg kg$^{-1}$ of body weight of D-Luciferin (Promega)[61]. Images

were analyzed using Perkin Elmer 3.2. In Vivo Imaging Software. Tumor size was also determined using a standard caliper and its volume calculated using the formula (width$^2$ length$^{-1}$) × 0.52, where width represents the shortest tumor dimension[62]. In the xenograft models, when HCT116 tumors reached ~100 mm$^3$ of volume, animals were randomly allocated into different groups and were treated 5 days a week with a single daily intraperitoneal injection of 10 mg kg$^{-1}$ nebivolol, 0.2 mg kg$^{-1}$ 5-fluorouracil (5FU) or 10 mg kg$^{-1}$ nebivolol combined with 0.2 mg kg$^{-1}$ 5FU. Likewise, the same procedure was followed for mice bearing MDA-MB-231 tumors that were treated with 10 mg kg$^{-1}$ nebivolol. In both HCT116 and MDA-MB-231, a 0.9% NaCl-treated group was included as a control. Following the ethical criteria established by our Institutional Review Board, the animals were sacrificed when the tumor volume reached ~ 2000 mm$^3$ and the tumor removed for further analysis. In the orthotopic model, mice were allocated into the control and nebivolol-treated groups when a stable luminescence signal of the implanted cells was attained and further treated as above indicated. Animals were euthanized after 35 days of treatment due to heavy tumor burden observed in the control group. Tumors formed and metastatic colonization were evaluated postmortem.

**Cell lines**. Cells were cultured in a humidified incubator at 37 °C with a controlled atmosphere of ambient air 10% CO$_2$. Human colorectal carcinoma HCT116, HCT116-Luc, and shIF1 HCT116 cells were grown in McCoy's 5A media supplemented with 10% fetal bovine serum (FBS). The HCT116-Luc cells expressed luciferin and were used to analyze tumor growth in vivo. Human breast MDA-MB-231, shIF1 MDA-MB-231, lung A549, and ovarian OVCAR 8 carcinoma cells; normal rat kidney (NRK) and breast Hs 578T normal cells were grown in Dulbecco's Modified Eagle's - Medium (DMEM) supplemented with 10% FBS. Mouse myoblasts C2C12 cells were grown in αMEM with 10% FBS and differentiated in αMEM containing 2% FBS and 100 nM insulin. Human neuroblastoma SH-SY5Y cells were grown in DMEM-F12 media supplemented with 10% FBS. Primary cortical neuronal cultures were obtained from E15 to E16 mouse embryos and plated at a density of $5 \times 10^5$ cells cm$^{-2}$ on poly-L-lysine and laminin-coated pretreated glass coverslips in Neurobasal medium supplemented with 2% B27, 1% glutamax (all from Invitrogen) and 100 mg ml$^{-1}$ penicillin–streptomycin. On the fifth day in vitro, half of the plating medium was removed from each well and replenished with BrainPhys medium (Stem cell Technologies) supplemented with 2% B27 and 100 mg ml$^{-1}$ penicillin–streptomycin[44]. HUVEC cells were kindly provided by Dr. Jaime Millán (CBMSO, Madrid) and grown on fibronectin-coated flasks (10 µg ml$^{-1}$) with EBM-2 medium supplemented with 2% FBS and endothelial cell growth factor (EGM-2) following the manufacturer's instructions (Lonza, Walkersville)[63].

**Cell transfection**. Cells were used at ~60% confluence. The NRK cell line was maintained in DMEM and transfected with pCMV-SPORT6-IF1, the phospho-deficient (S39A) or phosphomimetic (S39E) IF1 mutants[18]. Cells were harvested 24 h post-transfection and processed for 2D-gel electrophoresis.

**Isolation of mitochondria**. Cells were homogenized in a glass-Teflon homogenizer with seven volumes of hypotonic buffer (83 mM sucrose, 10 mM MOPS pH 7.2). After homogenization, the same volume of hypertonic buffer (250 mM sucrose, 30 mM MOPS pH 7.2) was added and nuclei and unbroken cells were removed by centrifugation at $1000 \times g$. Mitochondria were obtained by centrifugation at $12,000 \times g$ and washed in buffer A (320 mM sucrose, 1 mM ethylenediaminetetraacetic acid (EDTA), 10 mM Tris-HCl pH 7.4)[18,64].

**Drug library screening**. The effects of compounds from a 1018 FDA-Approved Drug Library (Selleckchem) on mitochondrial respiratory parameters of HCT116 cells were determined in XFe96 and XF24 Seahorse flux analyzers (Agilent Technologies) using 10 mM glucose, 1 mM pyruvate and 2 mM glutamine. A primary XFe96 screening to determine the 1 µM short-term effect (3 h) of the drugs on the basal and OSR was carried out on 35,000 cells. Compounds that negatively affected respiration by a factor of at least 40% were selected for XF24 secondary screening. For the respiration using palmitate as a substrate, cells were starved for 12 h in low-glucose DMEM (0.05 mM glucose, 1% FBS), and then changed to KHB media (111 mM NaCl, 4.7 mM KCl, 1.25 mM glutamine, 5 mM 4-(2-hydroxyethyl)-1-piperazineethanesulfonic acid (HEPES), pH 7.4). Bovine serum albumin (BSA)-conjugated palmitate (1 mM sodium palmitate, 0.17 mM BSA solution) was added as the main substrate. To assess OSR, maximum respiration, and non-mitochondrial-dependent oxygen consumption, respectively, 6 µM oligomycin (OL), 0.75 mM 2,4-dinitrophenol (DNP), and 1 µM rotenone plus 1 µM antimycin were used.

**Oxygen consumption in isolated liver mitochondria**. Mouse livers were homogenized with 4 ml g$^{-1}$ of cold homogenization buffer A (320 mM sucrose, 1 mM EDTA, 10 mM Tris-HCl pH 7.4) and centrifuged for 10 min at $1000 \times g$ at 4 °C. The resulting supernatant was centrifuged for 10 min at $8000 \times g$ at 4 °C to pellet the mitochondria[46]. The oxygen consumption rates were determined in a Clark-type electrode. Glutamate plus malate (10 mM) were used as respiratory substrates in the presence or absence of 0.5 mM adenosine diphosphate (ADP), 6 µM OL, 5 µM carbonyl cyanide-4-(trifluoromethoxy)phenylhydrazone (FCCP), and 1 µM antimycin A. The composition of the respiration buffer is 75 mM mannitol, 25 mM sucrose, 20 mM Tris–HCl, 5 mM phosphate, 0.3 mM ethyleneglycoltetraacetic acid (EGTA), 0.5 mM EDTA, 100 mM KCl, 0.1% BSA, pH 7.4[44].

**Determination of ATP synthase activities**. Digitonin-permeabilized HCT116 or MDA-MB-231 cells were used for determining the mitochondrial ATP production. Permeabilized cells were resuspended in respiration buffer (225 mM sucrose, 10 mM KCl, 5 mM MgCl$_2$, 0.05% w/v BSA, 10 mM potassium-phosphate buffer, 1 mM EGTA and 10 mM Tris-HCl; pH 7.4) and added to a luminometer plate-reader. ATP production was measured as luminescence production in respiration buffer containing 0.1 mM ADP, 5 mM succinate, 0.15 µM P$^1$,P$^5$-di(adenosine-5′) pentaphosphate, 0.25 mg ml$^{-1}$ of luciferin and 0.02 mg ml$^{-1}$ luciferase[65]. Relative light units were converted to ATP concentration using an ATP standard curve. Isolated mitochondria from HCT116 or MDA-MB-231 cells were used for the spectrophotometrical determination of ATP synthase hydrolytic activity following the changes in absorbance at 340 nm (A$_{340}$)[65]. Inhibition of both activities was accomplished by the addition of 30 µM OL.

**Mitochondrial enzyme activities and rates of glycolysis**. Isolated mitochondria from HCT116 or MDA-MB-231 cells were used for the spectrophotometric determination of complexes I, II, IV[66]. Complex I activity was measured at A$_{340}$ using 100 µg of mitochondria in 1 ml C1/C2 buffer (25 mM K$_2$HPO$_4$, 5 mM MgCl$_2$, 3 mM KCN and 2.5 mg ml$^{-1}$ BSA) containing 0.1 mM UQ$_1$, 0.1 M NADH and 1 mg ml$^{-1}$ antimycin A. Inhibition of the activity was accomplished by the addition of 1 µM rotenone. Complex II activity was measured at A$_{600}$ using 100 µg of mitochondria in 1 ml C1/C2 buffer (25 mM K$_2$HPO$_4$, 5 mM MgCl$_2$, 3 mM KCN and 2.5 mg ml$^{-1}$ BSA) containing 30 µM DCPIP, 1 µM rotenone, 1 µM antimycin A, 10 mM succinate and 6 mM phenazine methosulfate. Complex IV was measured at A$_{550}$ using 100 µg of mitochondria in 10 mM KH$_2$PO$_4$, pH 6.5, 0.25 M sucrose, and 1 mg ml$^{-1}$ BSA containing 10 µM reduced cytochrome c. Cytochrome c solution was freshly reduced by adding some crystals of sodium dithionite. Inhibition of the activity was accomplished by the addition of 240 µM KCN. To determine the rates of glycolysis, the initial rates of lactate production were determined by the enzymatic quantification of lactate concentrations in the culture medium. Culture medium was replaced by fresh medium supplemented with 1% FBS 1 h before the measurement. Samples (200 µl) of culture medium were taken at different intervals (0, 30, 60 and 90 min) and precipitated with 800 µl of cold perchloric acid, incubated on ice for 1 h and then centrifuged for 5 min, $11,000 \times g$ at 4 °C to obtain a protein-free supernatant. The supernatants were neutralized with 20% (w/v) KOH and centrifuged at $11,000 \times g$ and 4 °C for 5 min to sediment the KClO$_4$ salt. Lactate levels were determined spectrophotometrically by following the reduction of NAD+ at A$_{340}$ after the addition of 10 µl of LDH.

**Determination of the mitochondrial membrane potential**. 300,000 cells were incubated with 0.5 µM TMRM$^+$ in 300 µl of FACS (1 mM EDTA, 2% FBS in Phosphate Buffered Saline (PBS)) and processed for flow cytometry to determine the mitochondrial membrane potential with 1 µM nebivolol, 5 µM FCCP, 6 µM oligomycin or left untreated. For each analysis, 10,000 events were recorded[15]. Data were analyzed in FlowJo software v10.6.2.

**Cellular proliferation, cell-death assays, and ROS production**. Cellular proliferation was determined by the incorporation of 5-ethynyl-20deoxy-uridine (EdU) into cellular DNA using the Click-iT EdU Flow Cytometry Assay Kit (Thermo Fisher Scientific)[19] and using CellTrace$^{TM}$ Far Red (Thermo Fisher Scientific), following the manufacturer's instructions. Cell cycle was analyzed by flow cytometry. After treatment, cells were trypsinized, centrifuged at $1000 \times g$ for 5 min, collected and washed with ice-cold PBS. Cellular pellets were resuspended and fixed with cold 70% ethanol overnight. After another wash with PBS, the cell pellets were resuspended in 1 ml of staining solution containing propidium iodide (PI, 50 µg ml$^{-1}$). Finally, the cells were incubated at 37 °C for 30 min in the dark before analysis. For cell death assays, 50,000 cells/well were seeded and treated with 1 µM staurosporine (STS), 120 µM hydrogen peroxide (H$_2$O$_2$) or 1 µM tamoxifen as indicated. Cell death was determined by flow cytometry after staining with annexin V and 7-AAD (Annexin V Apoptosis Detection Kit I, BD Pharmingen$^{TM}$)[19]. The intracellular production of hydrogen peroxide was monitored by flow cytometry using 10 µM 6-carboxy-2′,7′-dichlorodihydrofluorescein diacetate (DCFH2-DA) (Thermo Fisher Scientific)[19]. Cells were analyzed in a BD FACScan. For each analysis, 10,000 events were recorded. Data were analyzed in FlowJo software v10.6.2.

**Protein extraction and western blot analysis**. Cells or isolated mitochondria were resuspended in lysis buffer containing 25 mM Hepes, 2.5 mM EDTA, 1% Triton X-100 supplemented with protease and phosphatase inhibitor cocktails. Tumor samples from humanely sacrificed mice were freeze-clamped in liquid nitrogen. Tumor proteins were extracted in a buffer containing 50 mM Tris-HCl pH 8.0, 1% NaCl, 1% Triton X-100, 1 mM dithiothreitol (DTT), 0.1% sodium dodecyl sulfate (SDS), 0.4 mM EDTA, supplemented with protease and phosphatase inhibitor cocktails. Lysates were clarified by centrifugation at $11,000 \times g$ for 15 min. The resulting supernatants were fractionated on SDS-PAGE and transferred onto polyvinylidene difluoride (PVDF) or nitrocellulose membranes for

immunoblot analysis. Protein concentrations were determined using Bradford reagent (Bio-Rad protein assay). The primary antibodies used were anti-human IF1 (clone 14/2, 1:100)[15], anti-β-F1-ATPase (clone 11/21-7A8, 1:1000)[67], anti-GAPDH (1:20,000, Abcam), anti-NDUFA9 (1:1000, Abcam), anti-NDUFS3 (1:1000, Abcam) anti-SDH-B (1:500, Invitrogen), anti-Core 2 (1:500, Abcam), anti-COX4 (1:1000, Abcam), anti-Hsp60 (Stressgene SPA-807, 1:2000), anti-phosphoserine (1:100, Sigma), anti-nitrotyrosine (1:1000, Abcam), anti-glutathione reductase (1:1000, Santa Cruz Biotechnology), anti-α-tubulin (1:1000, Sigma), anti-catalase (1:20,000, Sigma), anti-SOD2 (1:1000, Abcam) anti-PRx6 (1:1000, Abcam), anti-PRx3 (1:1000, Invitrogen), anti-G6PDH (1:1000, Cell signaling), anti-VDAC (1:500, Abcam). anti-ADRB1 (1:500, Santa Cruz Biotecnologies), anti-VEGF (1:1000, Abcam), anti-VEGFR2 (1:1000, Cell Signaling), anti-phosphoERK (1:500, Cell Signaling) and anti-ERK (1:500, Santa Cruz Biotechnologies). Peroxidase-conjugated anti-mouse or anti-rabbit IgGs (Promega, 1/3000) were diluted in 5% non-fat-dried milk in Tris Buffered Saline (TBS) with 1% Tween 20 and used as secondary antibodies. The Novex® ECL (Thermo Fisher Scientific) system was used to visualize the bands. The intensity of the bands was quantified using a GS-900™ Calibrated Densitometer (Bio-Rad) and ImageJ software.

**Immunofluorescence labeling**. Human umbilical vein endothelial cells were fixed in 4% paraformaldehyde, permeabilized with 0.1% Triton X-100 (Merck Millipore) and labeled with anti-VEGFR2 (1:1000, Cell Signaling). Nuclei were counter stained with 4′,6-diamidino-2-phenylindole (DAPI) reagent. Cellular fluorescence was analyzed by confocal microscopy in a Zeiss LSM 710 inverted confocal microscope. The images were processed with ImageJ software.

**RNA extraction and quantification**. RNA was extracted and purified from cells with Trizol reagent (Thermo Fisher Scientific), according to the manufacturer's instructions. Purified RNA was quantified with a Nanodrop Spectrophotometer (Thermo Fisher Scientific), and 1 μg was retrotranscribed into cDNA with the High-Capacity cDNA Reverse Transcription Kit (Thermo Fisher Scientific). IF1 mRNA levels were analyzed by real-time PCR with retrotranscribed cDNA, Fast SYBRMasterMix (Thermo Fisher Scientific), and ABI Prism 7900HT sequence detection system (Thermo Fisher Scientific) at the Genomics and Massive Sequencing Facility (CBMSO–UAM). Primers used to amplify the target genes were as follows: human IF1 (forward 5′-GGGCCTTCGGAAAGAGAG-3′, reverse 5′-TTCAAAGCTGCCAGTTGTTC-3′), and human β-actin (forward 5′-CCAAC CGCGAGAAGATGA-3′, reverse 5′-CCAGAGGCGTACAGGGATAG-3′). Standard curves with serial dilutions of pooled cDNA were used to assess amplification efficiency of the primers and to establish the dynamic range of cDNA concentration for amplification. The relative expression of the mRNAs was determined with the comparative ΔΔCt method with β-actin as control.

**2D-gels and blue native gel electrophoresis**. Isoelectrofocusing (IEF) was performed with 13-cm Immobiline DryStrips of 6–11L [linear] pH gradient using an Ettan IPGPhor3 IEF unit (GE Healthcare)[18]. In brief, 200 μg of cellular protein diluted in 250 μl of rehydration buffer (DeStreak Rehydration Solution, GE Healthcare) containing 0.5% of the corresponding IPG buffer (GE Healthcare) were loaded in the 13-cm strips. The equilibrated strips were transferred to the top of a 12% SDS polyacrylamide gel. Electrophoresis was carried out using a Protean II XI system (Bio-Rad) with constant current (30 mA/gel) at 4 °C for 3 h. Western blot analysis of the fractionated proteins was performed as described above. For Blue Native (BN) gels, mitochondrial pellets were suspended in 50 mM Tris-HCl pH 7.0 containing 1 M 6-aminohexanoic acid at a final concentration of 10 mg ml⁻¹. The membranes were solubilized by the addition of 10% digitonin (4:1 digitonin/ mitochondrial protein). 5% Serva Blue G dye in 1 M 6-aminohexanoic acid was added to the solubilized membranes. Native PAGE™ Novex® 3–12% Bis-Tris Protein Gels (Life Technologies) were loaded with 70 μg of mitochondrial protein. After fractionation, the gels were electroblotted onto PVDF membranes. Membranes were further processed for immunoblotting.

**Immunocapture and immunoprecipitation assays**. Respiratory Complex I was immunopurified from isolated mitochondria of HCT116 or MDA-MB-231 cells using a commercial kit (Abcam) according to the manufacturer's instructions. For IF1 immunoprecipitation, cells were lysed with 50 mM Tris-HCl, pH 6.0, 150 mM NaCl, 0.5% Nonidet P40 with cOmplete Mini, EDTA-free protease inhibitor cocktail and phosphatase inhibitor cocktail 2. Protein from cell lysates (400 mg) was incubated with 12 μg of the indicated antibody bound to EZ View Red Protein G Affinity Gel at 4 °C overnight. The beads were washed twice before complexes were eluted and fractionated on SDS-PAGE. After fractionation, the gels were electroblotted onto PVDF membranes. Membranes were further processed for immunoblotting.

**Protein identification by reverse phase-liquid chromatography-MS/MS**. A symmetrical gel was prepared to allocate one part to Coomassie staining and another to western blot. The band revealed by the antibody was used as a reference to locate and cut the gel band to perform the in-gel digestion. After drying, gel bands were destained in acetonitrile:water (ACN:H₂O, 1:1), were reduced and alkylated. In brief, disulfide bonds were reduced with 10 mM DTT for 1 h at 56 °C,

and the thiol groups were alkylated with 50 mM iodoacetamide for 1 h at room temperature in the dark. Later, the bands were digested in situ with sequencing grade trypsin (Promega) and chymotrypsin (Roche)[68]. Acetonitrile was pipetted out, and the gel pieces were dried in a SpeedVac. Half of the dried gel pieces were reswollen in 50 mM ammonium bicarbonate pH 8.8 with 12.5 ng μl⁻¹ trypsin for 1 h in an ice-bath and the other half in 100 mM Tris HCl, 10 mM CaCl₂ pH 8 with 25 ng μl⁻¹ for 1 h. The digestion buffer was removed, and gels were covered again with 50 mM NH₄HCO₃ (trypsin) or 100 mM Tris HCl, 10 mM CaCl₂ (chymotrypsin) and incubated at 37 °C for 12 h for trypsin and 25 °C for chymotrypsin. Digestion was stopped by the addition of 1% trifluoroacetic acid (TFA). Whole supernatants were dried down and then desalted onto ZipTip C18 Pipette tips (Millipore) until the mass spectrometric analysis.

To perform the reverse phase-liquid chromatography-MS/MS analysis, the desalted protein digest was dried, resuspended in 10 μl of 0.1% formic acid and analyzed by RP-LC-MS/MS in an Easy-nLC II system coupled to an ion trap LTQ-Orbitrap-Velos-Pro hybrid mass spectrometer (Thermo Scientific). The peptides were concentrated (online) by reverse phase chromatography using a 0.1 mm × 20 mm C18 RP precolumn (Thermo Scientific) and then separated using a 0.075 mm × 250 mm C18 RP column (Thermo Scientific) operating at 0.3 μl min⁻¹. Peptides were eluted using a 90-min dual gradient from 5 to 25% solvent B in 68 min followed by a gradient from 25 to 40% solvent B over 90 min (Solvent A: 0.1% formic acid in water, solvent B: 0.1% formic acid, 80% acetonitrile in water). ESI ionization was done using a Nano-bore emitters Stainless Steel ID 30 μm (Proxeon) interface. The Orbitrap resolution was set at 30,000. Peptide identification from raw data was carried out using PEAKS Studio 8.5 search engine (Bioinformatics Solutions Inc.)[69]. Database search was performed against uniprot-homo sapiens.fasta 12/03/2018 containing 71,790 sequences (decoy-fusion database). False discovery rates for peptide-spectrum matches were limited to 0.01. Only those proteins with at least two distinct peptides being discovered from LC/MS/MS analyses were considered reliably identified.

**Tumor analysis**. Breast and colon tumors were fixed in 4% paraformaldehyde (Merck) and included in OCT blocks. Frozen 15-μm sections were incubated with fluorescein-labeled GSL I-B4 (5 μg ml⁻¹, Vector Laboratories), anti-CD31 (1:200, BD Pharmingen), anti-Actin, α-Smooth Muscle-Cy3 (1:100, Sigma) and anti-Laminin (1:25, Sigma) to assess angiogenesis, anti-activated caspase-3 (1:200, Cell Signaling) to assess cell death and anti-Ki67 (1:250, Thermo Fisher Scientific) to assess proliferation. Nuclei were counter stained with DAPI (diamidino-2-feni-lindol) reagent. Cellular fluorescence was analyzed by confocal microscopy in a Nikon A1R+ microscope. The images were processed with ImageJ v1.46 software.

Aliquots of freeze-clamped tumor powder were used to measure total ATP content using the ATP Bioluminescence Assay Kit CLS II (Sigma-Aldrich) following the manufacturer's instructions.

**Statistics**. The results shown are means ± SEM. Statistical analysis was performed by Student's *t* test. Statistical tests were two-sided at the 5% level of significance. Statistical analyses were performed using Excel Microsoft 365 and GraphPad Prism 7. Survival curves were derived from Kaplan–Meier estimates and compared by log-rank test. Tests were calculated using the SPSS 24.0 software package (IBM).

**Reporting summary**. Further information on research design is available in the Nature Research Reporting Summary linked to this article.

## Data availability

The mass spectrometry proteomics data have been deposited to the ProteomeXchange Consortium via the PRIDE partner repository with the dataset identifier PXD019750. All the other data supporting the findings of this study are available within the article and its supplementary information files and from the corresponding author upon reasonable request. A reporting summary for this article is available as a Supplementary Information file. Source data are provided with this paper.

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

## Acknowledgements

We thank Dr. Lucía González Llorente (CBMSO) for kind assistance with flow cytometry. We acknowledge the help from Dr. A. Marina and L. Peláez from the Proteomic, and C. Sánchez from the Confocal Microscopy Facilities of the CBMSO. C.N.-T. was supported by a predoctoral fellowship from FPI-MINECO and Fondo Social Europeo. L.F. received support from the Ramón y Cajal Program (RyC-2013-13693). The work was supported by grants from MINECO (SAF2016-75916-R and PID2019-108674RB-100), CIBERER-ISCIII (CB06/07/0017) and Fundación Ramón Areces, Spain.

## Author contributions

C.N.-T., F.S., K.S., L.F. and C.N.d.A. did the research and analyzed data; M.G.d.C. and L.F. contributed to the design of the study; C.N.-T. and J.M.C. designed research, analyzed data, and wrote the paper. All the authors read, contributed and approved the final manuscript.

## Competing interests

The authors declare no competing interests.
