## [Peer Review File · Nature Communications]

Reviewers' comments:

Reviewer #1 (Remarks to the Author):

This manuscript examines the effects of nebivolol on OXPHOS. The drug was identified from a screen. Here are some suggestions to consider:

- Overall, the topic is of interest, but the paper only addresses several of the items superficially. See detailed comments below.
- A major question is whether the observed effects are truly beta1 adrenergic receptor (ADRB1) driven. While nebivolol is thought to be beta1-selective, most of these drugs can have other effects. Several questions should be carefully considered in this setting: 1) Are the observed effects due to ADRB1 or are they off-target effects? 2) Do the cells being tested here express ADRB1? Are the effects dependent on presence of ADRB1? Are the downstream effects on metabolism truly PKA dependent? Can ADRB2 or ADRB3 mediate similar effects?
- ADRBs are known to mediate a number of downstream cancer-promoting pathways. To what extent is the OXPHOS pathway particularly dominant/important for observed functional/biological effects, as compared to other effects (e.g., immune, angiogenesis, etc.)?
- Many stromal populations are known to express ADRBs. Are your observed effects in vivo cancer cell specific? This should be rigorously tested.
- Page 9. It is unclear why subcutaneous models were used, especially for a project where the microenvironment can affect metabolism. Orthotopic model(s) would be very important to test and present.
- Page 11. MDA-MB-231 is a triple-negative line; as such, it is not clear why tamoxifen would be used? It should not really be sensitive to tamoxifen.
- The Discussion of almost 5 pages is diffuse and a bit difficult to follow. It could be cut down substantially and made much more focused.

Reviewer #2 (Remarks to the Author):

In this report by Cristina Nuevo-Tapióles et al, the authors screened a FDA-approved library to identify drugs targeting the mitochondrial respiration for repurposing as putative anticancer treatment. From this screen, they obtained 13 hits, among which nebivolol; the adrenergic receptor

β -blocker was further characterized. The author claimed that the screen was selective to OSR; however, this is just a figure of speech, as not only the OSR was affected but the basal and maximal respiration suggesting a more unspecific inhibition. Comparison of basal and maximal respiration between control and treated samples would generate similar inhibition as in 1b. More importantly, the discrepancy between the in vitro data (reduction in respiration and ATP production but no cytotoxic nor cytostatic effect) and the in vivo data where the treatment show cytotoxicity is a clear demonstration that the mechanism is not clear. The described effect of nebivolol on the respiratory chain structure/function and IF1 phosphorylation status developed in fig 1-5 cannot explain by them self the results obtained in Figure 6 and 7. In the discussion the author raised the possibility of nebivolol affecting the tumor vasculature as an additional and perhaps synergistic effect but this would need to be demonstrated clear to make this publication fit for the journal. Surprisingly, the author did not raise nor test the possibility secondary metabolite of nebivolol could be the most effective compound.

Therefore, although this work address the fundamental and important quest for better cancer therapy and how to deliver it faster to the clinic, the quality of the data presented and the lack of mechanism of action reconciling the in vitro and in vivo observation cannot support publication at Nat. Comm at least in its present form.

As stated in the general comment, it is clear that the compound affect not only OSR but also basal and maximal respiration and this need to be more transparent in the manuscript.

The fact that blocking of the respiratory chain yet little ROS production is observed in vitro need to be explained. What is the effect on the membrane potential?

P value is needed for Figure 1b, 1c and 2a. Are these difference significant?

The quality of the BN WB does not allow following the author conclusion. Moreover, it seems there is opposite results for I+III2 when probed with NDUFA9 or core2. Far better BN Blot are needed.

What is the level of IF1 in control and treated cancer cell in vivo?

Figure 4f showed that knockdown of IF1 protect from NEB inhibition on the respiration.

An easy way to connect the in vitro work with the in vivo data would to test whether HCT116 or MDA-MB-231 KO for IF1 became resistant to NEB treatment.

Figure 6b, c and I, suggest that there is no benefit of the combination therapy. NEB alone as a stronger effect on tumor size and better survival. Sam for figure 7

Test the effect of NEB secondary metabolites in vitro for IF1 expression level, ROS production, oxidative stress marker, antioxidant enzyme expression and cell death.

Answers to Reviewers'

Reviewer #1 (Remarks to the Author):

This manuscript examines the effects of nebivolol on OXPHOS. The drug was identified from a screen. Here are some suggestions to consider: • Overall, the topic is of interest, but the paper only addresses several of the items superficially. See detailed comments below.

• A major question is whether the observed effects are truly beta1 adrenergic receptor (ADRB1) driven. While nebivolol is thought to be beta1-selective, most of these drugs can have other effects. Several questions should be carefully considered in this setting: 1) Are the observed effects due to ADRB1 or are they off-target effects? 2) Do the cells being tested here express ADRB1? Are the effects dependent on presence of ADRB1? Are the downstream effects on metabolism truly PKA dependent? Can ADRB2 or ADRB3 mediate similar effects?

We thank the reviewer for appreciating the interest of our contribution. In agreement with the reviewer's comment in the revised version of the manuscript we have addressed the experiments required to demonstrate that the effects of nebivolol are truly β 1-adrenergic receptor (ADRB1) driven.

Following the reviewer's suggestion, we have excluded possible off-target effects of nebivolol (NEB) on mitochondria by measuring the effect of NEB on mitochondrial respiration in isolated liver organelles. The results indicate that nebivolol has no effect on mitochondrial respiration (See new supplementary Fig. 1c), excluding possible off-target effects of NEB on mitochondria and further stressing the need of intact cells to promote its inhibitory action on mitochondrial respiration (See new Fig 1c-e). Next, we have analyzed the expression of ADRB1 in the cells utilized in our study. The results indicate that the cell types that respond to the action of nebivolol express ADRB1 whereas those cell types that do not respond to the drug do not express the ADRB1 receptor (See new Fig. 1j). In addition, incubation of the cells with other β 1 adrenergic antagonists also inhibit cellular respiration of cancer cells, albeit with less potency than NEB (See new Fig 1h). Moreover, β 2 and β 3 adrenergic specific inhibitors do not affect cellular respiration of cancer cells (See new Fig 1i) excluding that ADRB2 and ADRB3 receptors could mediate similar effects of those triggered by NEB. Overall, we are confident that the effects of NEB on mitochondrial respiration are not off-target effects and are mediated by β 1-adrenergic receptors.

On the other hand, we have already reported (see Fig. 4 in Cell Rep. 2015, 12, 2143) that incubation of HCT116 cells with H89 (a competitive inhibitor of PKA) or with PKI (a pseudosubstrate inhibitory peptide of PKA) inhibit the respiratory activity of cancer cells, thus supporting the implication of PKA or of a PKA-like activity in the downstream inhibitory effects of PKA on mitochondrial respiration. In addition, we have also reported that the *in vivo* administration of the β -agonist clenbuterol to mice activates mitochondrial oxidative phosphorylation in isolated heart mitochondria whereas administration of the β -antagonist propranolol has opposite effects (see Fig. 7 in Cell Rep. 2015, 12, 2143). Overall, our findings support both *in vitro* and *in vivo* the implication of the PKA/cAMP pathway in the regulation of mitochondrial respiration. Other laboratories also support the implication of PKA in regulating mitochondrial respiration (see refs: 23,24,38). A sentence in this regard has been included in the revised version of the manuscript (see end first paragraph on page 14).

• *ADRBs are known to mediate a number of downstream cancer-promoting pathways. To what extent is the OXPHOS pathway particularly dominant/important for observed functional/biological effects, as compared to other effects (e.g., immune, angiogenesis, etc.)?*

We thank the reviewer for this interesting question. Following the reviewer's indication in the revised version of the manuscript we have addressed the implication of targeting β 1-adrenergic receptors in the angiogenesis of the carcinomas. The results obtained demonstrate that nebivolol inhibits tumor angiogenesis by restraining the proliferation of endothelial cells further contributing to the arrest of tumor growth imposed by limiting OXPHOS. In brief, we show that endothelial cells express β 1-adrenergic receptors (See new Fig 7a) and that treatment of the cells with NEB inhibits proliferation (See new Fig 7d) by arresting cell cycle in G0/G1 phase (See new Fig 7b) mediated by the dephosphorylation of ERK (See new Fig 7c). These results nicely correlate with the *in vivo* observation that shows the inhibition of endothelial cell proliferation in the tumors as assessed by IB4 staining (See new Fig 6k and 7i) and a diminished expression of VEGFR2 (See new Fig 6l and 7j) in mice treated with nebivolol. Overall, we have further characterized the mechanism of action of nebivolol in restraining tumor angiogenesis, an additional hallmark of the cancer phenotype, that added to the inhibition of mitochondrial metabolism, emphasizes the relevance of β 1-adrenergic signaling in cancer therapy. We thank the reviewer for the relevance of this question.

• *Many stromal populations are known to express ADRBs. Are your observed effects in vivo cancer cell specific? This should be rigorously tested.*

As mentioned above, the effects of nebivolol are mediated by ADRB1 positive cells, both in cancer and in endothelial cells (see new Figures 1j and 7a, respectively). The inhibition of the β 1-adrenergic cascade causes the inhibition of OXPHOS in cancer cells and restrains proliferation of endothelial cells that represses tumor angiogenesis. It is likely that both pathways contribute to halt tumor growth.

• *Page 9. It is unclear why subcutaneous models were used, especially for a project where the microenvironment can affect metabolism. Orthotopic model(s) would be very important to test and present.*

We partially agree with the reviewer's comment. However, and since we observed no effect of nebivolol in cell death and proliferation in the *in vitro* experiments -which I should say was an initial discouraging finding- we decided to use the more rapid and affordable subcutaneous xenograft mouse model. The fact was that we succeeded in demonstrating the inhibition of tumor growth *in vivo* by NEB. In order to satisfy the reviewer's concern and to provide insight into the mechanism of action of nebivolol in cells of the microenvironment, rather than developing the orthotopic model, that will not add much and introduces more animal suffering, we decided to study directly the effect of the drug on HUVEC cells and the angiogenesis of the carcinomas. The results clearly indicate that the β 1-blocker nebivolol inhibits proliferation of endothelial cells preventing tumor angiogenesis. We thank the reviewer for raising this point.

• *Page 11. MDA-MB-231 is a triple-negative line; as such, it is not clear why tamoxifen would be used? It should not really be sensitive to tamoxifen.*

We agree with the reviewer's comment. However, this cancer cells lacks of specific treatment and there is a small percentage of hormone receptor negative breast cancer patients (5-10%) that by

unknown mechanism respond to tamoxifen administration. This is the reason why we decided to include a tamoxifen-treated group in the study. In any case, in the revised version of the manuscript we have omitted tamoxifen treatment.

- *The Discussion of almost 5 pages is diffuse and a bit difficult to follow. It could be cut down substantially and made much more focused.*

Thanks to the reviewer's comment we have substantially reduced the discussion (now approx. 3 ½ pages) and made it more focused.

Reviewer 2

- *In this report by Cristina Nuevo-Tapióles et al, the authors screened a FDA-approved library to identify drugs targeting the mitochondrial respiration for repurposing as putative anticancer treatment. From this screen, they obtained 13 hits, among which neбиволол; the adrenergic receptor β -blocker was further characterized. The author claimed that the screen was selective to OSR; however, this is just a figure of speech, as not only the OSR was affected but the basal and maximal respiration suggesting a more unspecific inhibition. Comparison of basal and maximal respiration between control and treated samples would generate similar inhibition as in 1b.*

We apologize to the reviewer for the misunderstanding in our explanation of the rationale we followed in the XFe96 screening. Certainly, the preliminary screening was focused in compounds that preferentially affected oligomycin sensitive respiration (OSR) because it represents an indirect determination of ATP synthase activity. Of course, the drugs affected the three parameters of mitochondrial respiration (basal, OSR and maximal) as we showed in the respiratory profiles of neбиволол. This is now explicitly stated in the revised version. However, and for the sake of simplicity, we represented the drugs' effect in OSR (Fig. 1a). To comply with the reviewer's request, and to prevent the misunderstanding, we have incorporated in new Table 1 the effect of the inhibitors on the basal respiration together with the effect on OSR, and indicated in the legend that the effect on the maximum respiration is of similar magnitude as that shown in other respiratory parameters. Exclusion of maximal respiration from Table 1 is to prevent the clumping of data. Moreover, in the revised version of the manuscript we have rewritten and simplified the sentences addressing the screening.

- *More importantly, the discrepancy between the in vitro data (reduction in respiration and ATP production but no cytotoxic nor cytostatic effect) and the in vivo data where the treatment show cytotoxicity is a clear demonstration that the mechanism is not clear. The described effect of neбиволол on the respiratory chain structure/function and IF1 phosphorylation status developed in fig 1-5 cannot explain by them self the results obtained in Figure 6 and 7. In the discussion the author raised the possibility of neбиволол affecting the tumor vasculature as an additional and perhaps synergistic effect but this would need to be demonstrated clear to make this publication fit for the journal.*

We agree with reviewer's comment. In the revised version of the manuscript we have provided evidence of the mechanism by which neбиволол arrests tumor growth reconciling *in vitro* and *in vivo* data. As already discussed under comments of R#1, in addition of the β 1-driven inhibitory

effects of nebivolol on OXPHOS in cancer cells we have now demonstrated that β 1-adrenergic receptors are involved in arresting angiogenesis of the tumors by inhibiting the proliferation of endothelial cells (please see detailed answers to comments 1 and 2 of R#1). We thank the reviewer for this comment because the experiments requested have helped to unveil an additional target of β 1-adrenergic receptors stressing the potential of nebivolol in cancer therapy.

•Surprisingly, the author did not raise nor test the possibility secondary metabolite of nebivolol could be the most effective compound.

In agreement with reviewer's argument, in the revised version of the manuscript we have addressed the potential cytotoxic/cytostatic effects of 4-OH-nebivolol, the major metabolite in the degradation pathway of NEB (Drug Metabolism and Disposition 2016, 44 (11) 1828-1831). We have found that 4-OH-nebivolol has no effect on mitochondrial respiration (See new Supplementary Fig 4a), cancer cell proliferation (See new Supplementary Fig 4b) nor cell death (See new Supplementary Fig 5); thus supporting the role of the β 1-antagonist in arresting cancer progression.

•Therefore, although this work address the fundamental and important quest for better cancer therapy and how to deliver it faster to the clinic, the quality of the data presented and the lack of mechanism of action reconciling the in vitro and in vivo observation cannot support publication at Nat. Comm at least in its present form.

We hope that with the new data incorporated in the revised manuscript, in which we document the clear implication of β 1-adrenergic receptor signaling in restraining OXPHOS in cancer cells and the mechanisms by which nebivolol inhibits angiogenesis by repressing endothelial cells proliferation, reconciling the *in vitro* and *in vivo* data the reviewer will support publication of our contribution at Nature Communications. As indicated by the reviewer, we are confident of the importance and clinical potential of our findings for cancer treatment.

Summary of last comments of R#2:

•As stated in the general comment, it is clear that the compound affect not only OSR but also basal and maximal respiration and this need to be more transparent in the manuscript.

According to reviewer's request, we have explicitly stated (see the beginning of 2nd and 3rd paragraphs of Results Section) that inhibitors and nebivolol specifically affect the basal, OSR and maximum respiration of cancer cells (as is shown in respiratory profiles (Fig. 1c-e)). Moreover, we have incorporated in Table 1 the effect of inhibitors on basal and indicated in the legends that the maximum respiration is similarly affected, despite having used the effect on OSR as and initial selection criterion (See new Table 1).

•The fact that blocking of the respiratory chain yet little ROS production is observed in vitro need to be explained. What is the effect on the membrane potential?

As requested by the reviewer, we have determined the mitochondrial membrane potential ($\Delta\Psi$ m) in nebivolol-treated HCT116 and MDA-MB-231 cells (See new Fig 2c). In agreement with the observed increase in ROS production, we observed a significant increase in $\Delta\Psi$ m in response to nebivolol treatment consistent with the inhibition of mitochondrial respiration by the drug. In fact, the nebivolol-driven increase in $\Delta\Psi$ m is similar to that exerted by oligomycin after inhibiting the

ATP synthase (See new Fig 2c). It is likely that the antioxidant capacity of cancer cells is able to buffer much of the ROS produced in these situations.

• *P value is needed for Figure 1b, 1c and 2a. Are these difference significant?*

As requested by the reviewer, the p value has been incorporated in the corresponding legends of the figures alluded. In all cases the differences are significant. Regarding Figure 2a, the plot (in the left) shows a representative assay of the effect of nebigivolol or of oligomycin in the kinetics of mitochondrial ATP production in permeabilized cells. The histograms (to the right) summarize all the data and show the statistical evaluation with an asterisk (*, $p < 0.01$).

• *The quality of the BN WB does not allow following the author conclusion. Moreover, it seems there is opposite results for I+III₂ when probed with NDUF9 or core2. Far better BN Blot are needed.*

We cannot agree with the comment that the quality of our blots is not good enough but partially understand the reviewer's concern about the interpretation of BN gels. That is the reason why we performed the experiments in three different mitochondrial preparations and quantified the amount of each complex and supercomplex according to its migration in the gels (See Fig. 3b). The slight differences alluded by the reviewer are vanished after the quantification of the three experiments as shown by the absence of statistical differences for supercomplex I+III₂ (See Fig. 3b). The blots included in Figure 3a are the ones more closely resembling all the migration patterns of respiratory complexes represented in Fig. 3b.

• *What is the level of IF1 in control and treated cancer cell in vivo?*

As shown in Fig. 6h and Fig. 7g of the previous version, now in Figs. 6g and supplementary Fig. 6b, the IF1 levels in cancer cells of nebigivolol-treated mice are 2-fold (HCT116 cells) and 1.3-fold (MDA-MB-231) higher than in NaCl-treated mice. Similar fold change differences in IF1 expression are observed when the cells are treated with nebigivolol *in vitro* (2.2 fold in HCT116 cells and 1.5 fold in MDA-MB-231 cells, Fig. 2e). In all cases the differences in IF1 expression triggered by nebigivolol are significant.

• *Figure 4f showed that knockdown of IF1 protect from NEB inhibition on the respiration.*

An easy way to connect the in vitro work with the in vivo data would to test whether HCT116 or MDA-MB-231 KO for IF1 became resistant to NEB treatment.

We cannot agree with the reviewer's suggestion because IF1 silencing did not prevent the inhibitory effect of nebigivolol on the maximum respiration. In fact, this was the finding that prompted our study on other complexes of the respiratory chain and led us to discover the effect of nebigivolol on the phosphorylation status and activity of complex I. Moreover, as we have shown with the new experiments requested, nebigivolol further prevents angiogenesis of the tumors. In other words, inhibition of β 1-adrenergic signaling has three targets to prevent tumor growth. Two of them are placed in mitochondrial OXPHOS, by arresting the activity of complex I and partially blocking the activity of the ATP synthase, and the third one is placed in endothelial cells of the microenvironment whose proliferation is partially inhibited compromising tumor vascularization.

• *Figure 6b, c and I, suggest that there is no benefit of the combination therapy. NEB alone as a stronger effect on tumor size and better survival. Sam for figure 7.*

In partial agreement with this reviewer's comment, the NEB+TAM treatment when compared to TAM or NEB treatments, respectively (old Fig. 7b), revealed no statistical significance. The main reasons for this lack of differences could stem from the low number of animals used in each group and that only a small percentage of triple negative breast cancer cells (MDA-MB-231) respond to tamoxifen treatment. As indicated previously under comment regarding page 11 of R#1, in the revised version of the manuscript we have eliminated the combined TAM+NEB treatment (see new Fig. 7e-f). However, we cannot agree with reviewer's comment regarding the combined NEB+5FU treatment because both in previous Fig. 6i,j (also in new Fig. 6i,j) there is a clear statistical improvement in mouse survival when the combined treatment is compared to CRL or 5FU treatments, respectively.

•*Test the effect of NEB secondary metabolites in vitro for IF1 expression level, ROS production, oxidative stress marker, antioxidant enzyme expression and cell death.*

Following the reviewer's suggestion, we have tested the effect of 4-OH-nebivolol, the major secondary metabolite of nebivolol, in mitochondrial respiration (See new Supplementary Fig 4a), cancer cell proliferation (See new Supplementary Fig 4b) and cell death (See new Supplementary Fig 5). We found no effect of 4-OH-nebivolol in any of these parameters, excluding the potential implication of the secondary metabolite in the *in vitro* effects of the drug in cancer cells. Since 4-OH-nebivolol had no effect on mitochondrial respiration we considered unnecessary to explore its effect on ROS production and in the antioxidant response.

Reviewers' comments:

Reviewer #1 (Remarks to the Author):

The revised paper has addressed several of the concerns. However, this reviewer does not find their arguments regarding the in vivo studies persuasive. An orthotopic model would certainly be more convincing than the current set of in vivo experiments.

Moreover, the added endothelial experiments are quite cursory and do not really provide convincing testing of angiogenesis.

Reviewer #2 (Remarks to the Author):

The authors have replied to all my comments and I do not have further comments. White the added data the story is more complete and better fit for acceptance.

Reviewer #3 (Remarks to the Author):

The study from Nuevo-Tapioles and colleagues investigates the role of β 1-adrenergic inhibition on mitochondrial activity and angiogenesis during tumor growth. The major findings are that:

The β 1-blocker nebivolol specifically hinders oxidative phosphorylation in cancer cells by concertedly inhibiting Complex I and ATP synthase activities.

Nebivolol also arrests tumor angiogenesis by arresting endothelial cell

36 proliferation in G0/G1 phase of the cell cycle.

The manuscript is well written and the experiments are well designed. This study contains some intriguing elements. However, the effect on tumor angiogenesis requires some further analysis.

Major:

The analysis of vessel density in tumors (Figures 6 and 7) was carried by using a single marker/readout, that is 4B1 staining. 4B1 staining is supposedly visualising only a subset of intratumoral blood vessels. Other reports show that 4B1 binds to laminin a basement membrane protein in blood vessels which is of the absent in tumor blood vessels, rather than to the endothelial cells. Hence, 4B1 is not suitable as a single readout to analyse the entire tumor vasculature. I suggest to perform an additional CD31 (endothelial cell) staining and maybe even a staining for the basement membrane (laminin) or pericytes (alpha-SMA).

Along this lines, this analysis should be complemented by functional vascular readouts, e.g. FITC-Lectin injection (vessel perfusion) or stainings that visualise tumor hypoxia (Hypoxyprobe, GLUT1, CAIX) to asses the ability of the blood vessels to deliver oxygen to the tumor.

The authors show decreased VEGFR2 expression after Nebivolol treatment. Does this reflect a decrease in the number of (VEGFR2-expressing) blood vessels or a decrease of VEGFR2 expression on the surface of endothelial cells. Here a VEGFR2 staining on tumor sections as well as in vitro treated endothelial cells could help to further dissect this out.

Finally, it would be interesting to know if and how Nebivolol affect the metabolism of endothelial cells. This would be even more interesting to know since endothelial cells usually depend on anaerobic glycolysis.

Answers to reviewers

Reviewer #1 (Remarks to the Author):

Q1: *The revised paper has addressed several of the concerns. However, this reviewer does not find their arguments regarding the in vivo studies persuasive. An orthotopic model would certainly be more convincing than the current set of in vivo experiments.*

A1: We thank the reviewer for appreciating the improvement of the manuscript. Following the reviewer's indication, we have now performed an orthotopic colon cancer model to further address the role of nebivolol *in vivo*. We have followed the growth of tumors by measuring the increase in luminescence and we have demonstrated that nebivolol significantly diminished tumor growth (New Fig. 7a,b). Furthermore, nebivolol-treated mice developed less and smaller tumors (New Fig. 7c), less micrometastasis and less blood vessels (New Fig. 7d) when compared to controls, further emphasizing the benefit of nebivolol treatment in the colon microenvironment. These results have been incorporated in the Results section of the revised version of the manuscript (See page 13).

Q2: *Moreover, the added endothelial experiments are quite cursory and do not really provide convincing testing of angiogenesis.*

A2: To satisfy the reviewers' concern, we have followed the approaches suggested by Reviewer 3 to assess the role of nebivolol in tumor angiogenesis. In the revised version of the manuscript, tumor angiogenesis has been evaluated by the inclusion of the following additional markers: CD31 of endothelial cells (New Fig. 5k and Fig. 8i); laminin (New Fig. 5l and Fig. 8j), marker of the basement membrane; and α -SMA (New Fig. 5m and Fig. 8k), marker of pericytes. The expression of these markers was also significantly reduced in tumor sections of colon and breast cancer specimens when mice were treated with nebivolol, further supporting that nebivolol inhibits tumor angiogenesis. Moreover, we have also demonstrated that the decrease in VEGFR2 in carcinomas in response to nebivolol treatment results from a diminished tumor angiogenesis due to the arrest of the proliferation of endothelial cells (New Fig. 6c-d). The arrest of proliferation of endothelial cells most likely results from the nebivolol-mediated inhibition of glycolysis (New Fig. 6f), which is a prime pathway for cellular proliferation.

Reviewer #2 (Remarks to the Author):

Q1: *The authors have replied to all my comments and I do not have further comments. White the added data the story is more complete and better fit for acceptance.*

A1. We thank the reviewer for appreciating the improvement of our work.

Reviewer #3 (Remarks to the Author):

Q1: *The study from Nuevo-Tapioles and colleagues investigates the role of β 1-adrenergic inhibition on mitochondrial activity and angiogenesis during tumor growth. The major findings are that:*

The β 1-blocker nebivolol specifically hinders oxidative phosphorylation in cancer cells by concertedly inhibiting Complex I and ATP synthases activities.

Nebivolol also arrests tumor angiogenesis by arresting endothelial cell proliferation in G0/G1 phase of the cell cycle.

The manuscript is well written and the experiments are well designed. This study contains some intriguing elements. However, the effect on tumor angiogenesis requires some further analysis.

A 1: We thank the reviewer for supporting and appreciating our work.

Major:

Q2: *The analysis of vessel density in tumors (Figures 6 and 7) was carried by using a single marker/readout, that is 4B1 staining. 4B1 staining is supposedly visualizing only a subset of intratumoral blood vessels. Other reports show that 4B1 binds to laminin a basement membrane protein in blood vessels which is of the absent in tumor blood vessels, rather than to the endothelial cells. Hence, 4B1 is not suitable as a single readout to analyse the entire tumor vasculature. I suggest to perform an additional CD31 (endothelial cell) staining and maybe even a staining for the basement membrane (laminin) or pericytes (alpha-SMA). Along this lines, this analysis should be complemented by functional vascular readouts, e.g. FITC-Lectin injection (vessel perfusion) or stainings that visualize tumor hypoxia (Hypoxyprobe, GLUT1, CAIX) to assess the ability of the blood vessels to deliver oxygen to the tumor.*

A2: Following the reviewer's indications, we have performed the staining of the tumor sections from colon and breast carcinomas with CD31, laminin and α -SMA. The staining of endothelial cells (CD31 staining, New Fig. 5k and Fig. 8i), laminin staining (basement membrane, New Fig. 5l and Fig. 8j) and α -SMA (pericytes, New Fig. 5m and Fig. 8k) also support a significant reduction of angiogenesis in carcinomas of nebigivolol-treated mice. Moreover, additional results that stress the inhibition of angiogenesis by nebigivolol in the carcinomas stem from the effect of the drug in limiting the proliferation of endothelial cells, rather than by affecting VEGFR2 expression (New Fig. 6a. and Fig. 8l). This effect of nebigivolol is most likely mediated by its profound inhibitory effect in the flux of glycolysis (New Fig. 6f), a prominent pathway required for cellular proliferation (See answers to Q3 and Q4 of R#3).

Despite the evidences described above for the nebigivolol-mediated inhibition of angiogenesis in colon and breast carcinomas, and due to the request of developing an orthotopic model (see R#1), we should mention that we have observed that nebigivolol reduces the number of blood vessels in carcinomas and areas of micrometastasis in the orthotopic model (New Fig. 7d, black arrows). In any case, in this model we wanted to determine vessel perfusion by FITC-Lectin injection to assess the functional vascular readouts of the carcinomas. The FITC-lectin was administered to mice and the tumors removed and fixed. Unfortunately, this stage of project development coincided with the outbreak of Covid-19 pandemic in Spain. Since then, we are experiencing a severe home-confinement that impedes us to reach the lab to finalize the experiments. Since this situation is going to last for much longer than expected, we have decided to proceed with manuscript submission. We consider that the evidences provided regarding the negative effect of nebigivolol in angiogenesis are satisfactory and, on the other hand, it could accelerate the translation of nebigivolol into cancer patients.

Q3: *The authors show decreased VEGFR2 expression after Nebigivolol treatment. Does this reflect a decrease in the number of (VEGFR2-expressing) blood vessels or a decrease of VEGFR2 expression on the surface of endothelial cells. Here a VEGFR2 staining on tumor sections as well as in vitro treated endothelial cells could help to further dissect this out.*

A3: Following the reviewer's requests we have performed VEGFR2 staining in control and nebigivolol-treated HUVEC cells. We show that nebigivolol treatment does not affect the

expression of VEGFR2 in endothelial cells (New Fig. 6b). In contrast, nebivolol significantly diminished the number of endothelial cells as assessed by immunofluorescence and flow cytometry (New Fig. 6c, d). Overall, these results indicate that the reduction of VEGFR2 expression in the carcinomas is not the result of decreased VEGFR2 expression on the surface of endothelial cells but results from a reduction in the proliferation of endothelial cells.

Q4: *Finally, it would be interesting to know if and how Nebivolol affect the metabolism of endothelial cells. This would be even more interesting to know since endothelial cells usually depend on anaerobic glycolysis.*

A4: To satisfy the reviewer's suggestion, we have tested the effect of nebivolol on mitochondrial respiration and glycolytic flux in endothelial cells. We found that nebivolol does not affect mitochondrial respiration (See Supplementary Fig. 6a). In contrast, nebivolol profoundly inhibits aerobic glycolysis of endothelial cells (New Fig. 6f), nicely correlating with the arrest of cellular proliferation (Fig. 6d) of the cells at G0/G1 (Fig. 6g, Supplementary Fig. 6b), due to the relevance of glycolysis in endothelial cell metabolism and in cellular proliferation. These results have been added in the new version of the manuscript (See pages 12 and 13).

We thank the reviewer for her/his suggestions that have improved our contribution on the mechanism of action of nebivolol in endothelial cells.

REVIEWERS' COMMENTS:

Reviewer#1:

My concerns have been addressed

Reviewer#2:

The authors have conducted the requested analysis, revised the manuscript and sufficiently addressed my questions. Congratulations to a nice piece of work

ANSWERS TO REVIEWERS' COMMENTS:

Reviewer#1:

My concerns have been addressed

We thank the reviewer for his/her comment.

Reviewer#2:

The authors have conducted the requested analysis, revised the manuscript and sufficiently addressed my questions. Congratulations to a nice piece of work

We thank the reviewer for his/her encouraging comment.